# Effective drug combinations in breast, colon and pancreatic cancer cells

Patricia Jaaks[1,8], Elizabeth A. Coker[1,8], Daniel J. Vis[2,7], Olivia Edwards[1], Emma F. Carpenter[1], Simonetta M. Leto[3], Lisa Dwane[1], Francesco Sassi[3], Howard Lightfoot[1], Syd Barthorpe[1], Dieudonne van der Meer[1], Wanjuan Yang[1], Alexandra Beck[1], Tatiana Mironenko[1], Caitlin Hall[1], James Hall[1], Iman Mali[1], Laura Richardson[1], Charlotte Tolley[1], James Morris[1], Frances Thomas[1], Ermira Lleshi[1], Nanne Aben[2], Cyril H. Benes[4], Andrea Bertotti[3,5], Livio Trusolino[3,5], Lodewyk Wessels[2,6,7] & Mathew J. Garnett[1✉]

Combinations of anti-cancer drugs can overcome resistance and provide new treatments[1,2]. The number of possible drug combinations vastly exceeds what could be tested clinically. Efforts to systematically identify active combinations and the tissues and molecular contexts in which they are most effective could accelerate the development of combination treatments. Here we evaluate the potency and efficacy of 2,025 clinically relevant two-drug combinations, generating a dataset encompassing 125 molecularly characterized breast, colorectal and pancreatic cancer cell lines. We show that synergy between drugs is rare and highly context-dependent, and that combinations of targeted agents are most likely to be synergistic. We incorporate multi-omic molecular features to identify combination biomarkers and specify synergistic drug combinations and their active contexts, including in basal-like breast cancer, and microsatellite-stable or *KRAS*-mutant colon cancer. Our results show that irinotecan and CHEK1 inhibition have synergistic effects in microsatellite-stable or *KRAS–TP53* double-mutant colon cancer cells, leading to apoptosis and suppression of tumour xenograft growth. This study identifies clinically relevant effective drug combinations in distinct molecular subpopulations and is a resource to guide rational efforts to develop combinatorial drug treatments.

Single-agent targeted therapies for patients with molecularly defined tumours are transforming cancer treatment. Nonetheless, many patients still lack effective treatments and pre-existing or acquired resistance limits the clinical benefit of even the most advanced medicines[2]. Empirically developed combinations of chemotherapy drugs are used to treat cancer patients[1]. Combination therapies using targeted anti-cancer agents have the potential to overcome resistance, enhance the response to existing drugs, reduce dose-limiting single agent toxicity and expand the range of treatments[2], as exemplified by triple combination therapy for patients with BRAF-mutant colorectal cancer[3]. However, our ability to predict effective combinations is limited[4]. Molecularly annotated cancer cell line panels[5] are increasingly being used to identify active drug combinations[4,6–8]. Studies performed so far have limitations, including testing relatively few combinations, using few molecularly targeted drugs, or using a limited number and sub-optimal range of drug concentrations. Furthermore, previous studies have employed a limited number of cell lines[4] (a maximum of 85), making it difficult to link combination activity and molecular context.

## Drug combination screens in cancer cells

To systematically identify active drug combinations, we used the Genomics of Drug Sensitivity in Cancer (GDSC) cell line screening platform[5] to measure the effects of 2,025 pairwise drug combinations (Supplementary Table 1) in 125 cell lines (Supplementary Table 2), including breast ($n = 51$), colorectal ($n = 45$; hereafter referred to as colon) and pancreatic ($n = 29$) cancer (Fig. 1a). We produced 296,707 drug combination viability measurements for 108,259 combination–cell line pairs, making it the second largest drug combination dataset by number of combinations and experiments, with the largest number of cell lines tested[4,6–8].

Each cell line has mutation, copy number alteration, methylation and gene expression data available (Extended Data Fig. 1a, Supplementary Table 2). We selected drugs for each tissue including chemotherapeutics and targeted agents approved by the United States Food and Drug Administration (FDA), drugs in clinical development and investigational compounds (Extended Data Fig. 1b, c). We enriched for drugs against key targets and pathways ($n = 20$), such as ERBB2 inhibitors in breast

[1]Wellcome Sanger Institute, Cambridge, UK. [2]Division of Molecular Carcinogenesis, The Netherlands Cancer Institute, Amsterdam, The Netherlands. [3]Candiolo Cancer Institute, FPO–IRCCS, Turin, Italy. [4]Massachusetts General Hospital, Harvard Medical School, Boston, MA, USA. [5]Department of Oncology, University of Torino School of Medicine, Turin, Italy. [6]Department of EEMCS, Delft University of Technology, Delft, The Netherlands. [7]Oncode Institute, Amsterdam, The Netherlands. [8]These authors contributed equally: Patricia Jaaks, Elizabeth A. Coker. ✉e-mail: mg12@sanger.ac.uk

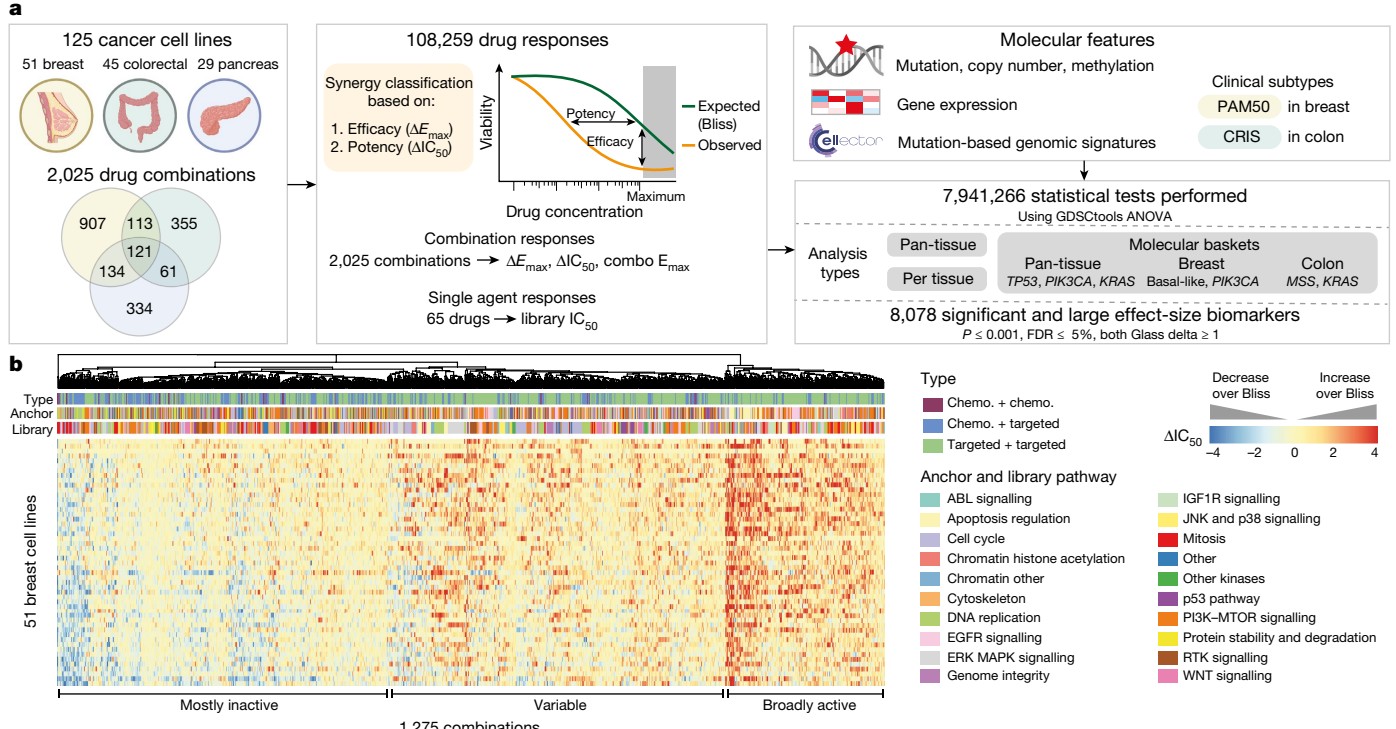

**Fig. 1 | A large-scale drug combination screen. a**, 2,025 drug combinations were screened in breast, colon and pancreas cancer cell lines ($n = 125$). Synergy was evaluated on the basis of efficacy ($\Delta E_{max}$) and potency ($\Delta IC_{50}$) for 108,259 drug responses and integrated with cell line molecular features to identify biomarkers. **b**, Heat map of $\Delta IC_{50}$ values for 1,275 combinations in 51 breast cancer cell lines: clustering by combination and annotation by combination type, anchor and library pathway. $\Delta IC_{50}$ limits are clipped to −4 and 4, rows are sorted by conditional mean $\Delta IC_{50}$ on cell line identity. Chemo., chemotherapeutic agent.

and drugs targeting MAPK signalling in colon and pancreas (Extended Data Fig. 1d, Supplementary Table 1). We tested 121 combinations in all three tissues.

To screen efficiently we used a 2 × 7 concentration matrix, or 'anchored' approach. We screened each anchor compound at two optimised concentrations and a discontinuous 1,000-fold (7-point) dose–response curve of the library compound (Extended Data Fig. 1e). Single-agent and combination viability measurements were fitted per cell line and multiple parameters derived including: (1) anchor viability effect, (2) library and combination viability effect at the highest-used library concentration (library $E_{max}$ and combination $E_{max}$, respectively), and (3) the estimated library drug concentration producing a 50% viability reduction (IC_{50}) for the library and combination (Extended Data Fig. 1f). We compared observed combination response of cells to the Bliss independence-predicted response[9] based on monotherapy activity, and classified drug combinations on the basis of shifts beyond Bliss in potency ($\Delta IC_{50}$; that is, increased sensitivity) or efficacy[10] ($\Delta E_{max}$; that is, reduced cell viability) (Extended Data Fig. 1g). We classified combination–cell line pairs as synergistic if, at either anchor concentration, combination IC_{50} or $E_{max}$ was reduced eightfold or 20% viability over Bliss, respectively (Extended Data Fig. 1h). Stringent quality control was applied to all screening data: technical and biological replicates were highly correlated and library IC_{50} values were highly correlated with IC_{50} values from independent screens (Extended Data Fig. 2a–e, Supplementary Table 2).

Dimensional reduction (using *t*-distributed stochastic neighbour embedding (*t*-SNE)) on the 121 pan-tissue combinations showed moderate mixing of cell lines by tissue, indicating that tissue has some effect on combination response, but is not on its own a major driver of variation (Extended Data Fig. 2f), as previously described by others[8]. Clustering by $\Delta IC_{50}$ for all tissue-specific and pan-tissue combinations, we observed

that combinations fall into three major groups: (1) broadly active, (2) minimally active, and (3) variable activity (Fig. 1b, Extended Data Fig. 2g–i). All data are available for download or exploration through GDSC Combinations, https://gdsc-combinations.depmap.sanger.ac.uk/.

## The landscape of drug interactions

Overall, 5.2% of the 108,259 combination–cell line pairs showed synergy, with the highest rate in pancreas (7.2%), then colon (5.4%) and breast (4.4%). Only 27.5% of synergistic combination–cell line pairs were observed at both high and low anchor concentrations, suggesting that synergy is detected within a specific range of concentrations (Fig. 2a) and pointing to the utility of aggregating synergy calls from both anchor concentrations. Synergy occurred most frequently in a background of weak to moderate single-agent activity, enhancing existing drug responses (Extended Data Fig. 3a). 54.9% of synergistic measurements affected either efficacy (22.2%) or potency (32.7%), whereas 45.1% affected both, indicating that these two metrics describe complementary responses (Fig. 2b). Although synergy overall was rare, most frequently observed in less than 3 cell lines per tissue, 192 combination–tissue pairs (7.8%; 60 breast, 52 colon and 80 pancreas) were synergistic in at least 20% of cell lines from their respective tissue (Supplementary Table 1). The relative frequency and context specificity of synergy was retained independent of the synergy threshold applied (Extended Data Fig. 3b).

Rescreening a subset of 45–59 frequently synergistic combinations (51 breast, 45 colon and 59 pancreas) in 30 cell lines per tissue resulted in a validation dataset of 4,881 combination–cell line pairs. Primary and validation datasets correlated well for single-agent and combination response metrics ($r = 0.69$–$0.84$, all $P < 0.001$), and synergy classifications were consistent for all tissues (*F*-score: 0.62–0.7; recall: 0.61–0.76;

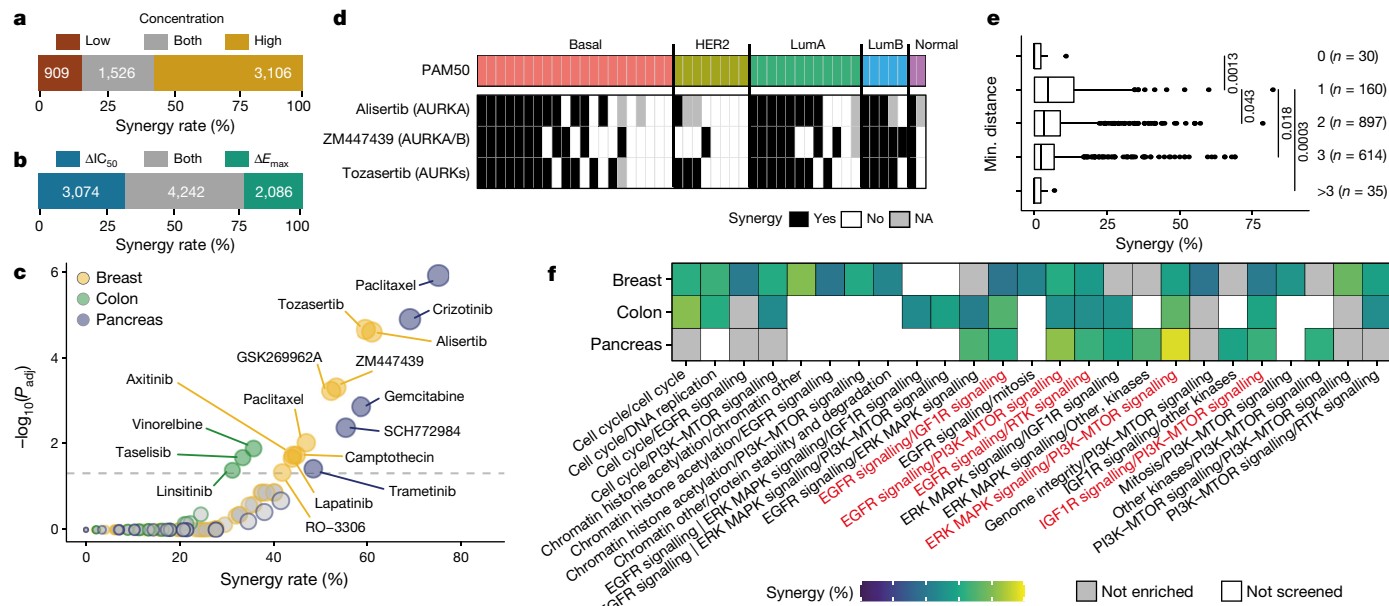

**Fig. 2 | Synergy is rare and highly context-dependent. a**, Overlap of synergy identified at two anchor concentrations. $n = 5,541$ synergistic combination-cell line pairs. **b**, Synergy calls by $\Delta IC_{50}$ and $\Delta E_{max}$ are complementary. $n = 9,402$ synergistic measurements. **c**, Navitoclax combination partners and tissue-specific enrichment in synergy (hypergeometric test). Enriched navitoclax partners ($P \le 0.005$, FDR $\le 5\%$) are labelled. **d**, Combinations of navitoclax with AURK inhibitors are frequently synergistic in breast cancer cell lines, with exception of HER2 cells. Binary synergy for navitoclax (anchor) paired with three AURK inhibitors with indicated specificity across PAM50 subtypes. **e**, Synergy rates for 1,736 combinations of two targeted drugs by minimum (min.) network distance between drug targets. Two-sided Student's *t*-test. **f**, Twenty-four unique target pathway pairs enriched in synergy in at least one tissue (136 unique pairs tested; hypergeometric test, $P \le 0.005$, FDR $\le 5\%$). Red denotes pathway pairs that are enriched in all three tissues.

precision rate: 0.56–77; original screening set defined as 'positive') (Extended Data Fig. 3c–e). Differences in classifications were frequently borderline cases and larger synergy effects were most reproducible (Extended Data Fig. 3f).

Of the 65 drugs in our screen, 10 were chemotherapeutic agents and 31.2% of all combinations contained one (28.7%) or two (2.5%) chemotherapeutic agents. Chemotherapeutic–chemotherapeutic combinations had lower $\Delta IC_{50}$ and $\Delta E_{max}$ shifts than combinations with targeted compounds (Extended Data Fig. 3g), resulting in low synergy rates (21 out of 2,337 combination–cell line pairs (0.9%)) compared with 3.2% and 6.1% for chemotherapeutic–targeted and targeted–targeted combinations, respectively.

We observed significant enrichment of synergy when chemotherapeutics were paired with drugs targeting apoptotic signalling and cell cycle inhibitors (hypergeometric test, adjusted $P \le 0.05$). More than 76% of synergies between chemotherapeutics and cell cycle inhibitors per tissue occurred with AZD7762 (CHEK1/2), pairing well with 5-fluorouracil, gemcitabine, cisplatin and irinotecan (Extended Data Fig. 3h).

Combinations with the apoptosis regulator navitoclax (an inhibitor of BCL-2, BCL-XL and BCL-W) comprised 25.4% of all synergistic combination–cell line pairs (1,418 out of 5,580), despite only representing 5.4% of combinations (109 out of 2,025). Seventy-eight per cent (137 out of 175) of combination–tissue pairs with navitoclax were synergistic in at least three cell lines (average per combination–tissue: 19.6% synergy). Navitoclax showed a synergy rate of more than 50% when combined with TOP1 inhibitors (irinotecan in pancreas and camptothecin in breast and colon) or microtubule stabilisers (docetaxel or paclitaxel) and destabilisers (vinorelbine) (Extended Data Fig. 3i). Targeted drugs that were synergistic with navitoclax were mostly tissue-specific (Fig. 2c). Navitoclax had particularly high synergy rates in breast when combined with either of the three aurora kinase (AURK) inhibitors alisertib, tozasertib or ZM447439 (61%, 60% and 53%, respectively). Navitoclax with alisertib and ZM447439 were

tested and had reproducible synergy in the validation screen (94% and 88%, respectively). Notably, synergy was frequently observed for at least two out of three navitoclax–AURKi combinations in all PAM50 subtypes (63% (12 out of 19 cell lines) in basal-like, 73% (8 out of 11) in LumA and 75% (3 out of 4) in LumB), with the exception of HER2 cell lines (17%, 1 out of 6 cell lines) (Fig. 2d). These data support ongoing efforts to use combinations suppressing anti-apoptotic adaptation in cancer[11], but indicate that in defined cancer types, pairing with specific targeted drugs is most likely to be effective.

For the 67% of combinations involving two targeted compounds, we investigated pathway relationships between drug targets and their synergy rate. We overlaid drug targets for all 57 targeted compounds onto a protein–protein interactome of 14,431 protein nodes and 110,118 edges based on the IntAct[12] database, filtered to unique human protein–protein interactions (confidence threshold: 0.5). We calculated the shortest finite network distance between the drug target nodes and found that on average combinations whose targets are between one and two nodes away from each other yield the most synergy (Fig. 2e), as reported previously[4], which indicates that there is an optimal average target distance to induce synergy.

To further understand how targeted–targeted drug combinations work at the pathway level, we grouped combinations into unique curated pathway pairs by tissue on the basis of the nominal therapeutic target(s) of each drug (excluding navitoclax combinations, as discussed separately, Supplementary Table 1). Eighteen per cent (25 out of 136) of pathway pairs were significantly enriched in synergy in at least one tissue (hypergeometric test, $P \le 0.005$, false discovery rate (FDR) $\le 5\%$) (Fig. 2f). Five pathway pairs were enriched in synergy in all three tissues, including dual targeting of receptor tyrosine kinase (RTK) signalling, targeting of RTK signalling with downstream PI3K or MAPK pathways, or targeting of PI3K and MAPK pathways. Taking PI3K and MAPK pathways as examples, we examined inter-pathway versus intra-pathway combinations. Inter-pathway targeting of PI3K

and MAPK pathways resulted in 2–5 times higher synergy rates than the tissue average: breast: 10.1% (versus 4.4%), colon: 17.1% (versus 5.4%) and pancreas: 36.4% (versus 7.2%). Conversely, intra-pathway targeting of the MAPK or PI3K pathways showed below or close to average rates of synergy (MAPK combinations: 2.2%; PI3K combinations: 6.7%; all combinations: 5.2%) (Extended Data Fig. 3j). Despite low intra-PI3K pathway synergy, we detected a distinct combination-specific synergy of MK-2206 (AKT1/2) combined with MTOR inhibitors OSI-027 or AZD8055 in breast (24% and 25% synergy rates, respectively), particularly in the PAM50 subtype HER2 (50–75%; 4 or 6 out of 8 cell lines) (Extended Data Fig. 3k). This combination may work by inhibiting feedback activation of AKT in HER2 breast cancer due to mTOR inhibitor-induced activation of PI3K signalling[13]. Numerous MAPK and PI3K inter-pathway combinations are currently in clinical trials, and preliminary data suggest that intermittent drug administration, isoform-selective PI3K inhibitors and site-specific delivery of drugs could maximise clinical activity while increasing tolerability[14].

Overall, we find that drug synergy, on the basis of complementary potency and efficacy metrics, is rare but frequent for a subset of combinations. Synergy most frequently enhances weak to modest single-agent drug response and rates are highest for combinations of targeted compounds. Combinations containing the apoptosis regulator navitoclax show high synergy rates. Whereas some trends are universal across all three tissues, such as high synergy rates for inter-pathway targeting of MAPK and PI3K pathways, other synergistic effects are tissue-specific, including navitoclax plus AURK inhibition in breast.

## Combination response molecular markers

Associations between molecular features and drug response, referred to here as biomarkers, can provide insights into the cellular behaviour that dictates response to drug treatment and can inform clinical development of therapies. We used GDSCTools[15] to identify multi-omics biomarkers of combination ($\Delta IC_{50}$, $\Delta E_{max}$ and combination $E_{max}$) and single-agent ($IC_{50}$) responses. Multiple analyses were performed, grouping cell lines in molecular contexts, including pan-tissue (3 tissues), individual tissues and seven molecular 'baskets' representing specific molecular subgroups (*TP53*, *KRAS*, *PIK3CA*, breast *PIK3CA*, basal-like breast cancers, colon *KRAS* and colon microsatellite-stable (MSS)) (Fig. 1a). *TP53*, *KRAS* and *PIK3CA* are the most frequently mutated genes across the cell lines, and the intra-tissue molecular baskets represent cancers with unmet clinical need.

We performed 7,941,266 analysis of variance (ANOVA) tests, out of which 8,078 associations were significant and of large effect size ($P \leq 0.001$, FDR $\leq 5\%$, Glass deltas for positive and negative populations both $\geq 1$) (Fig. 1a, Supplementary Table 3). Biomarkers of all feature types were found for all drug response inputs, except for CRIS and PAM50 classifications (Extended Data Fig. 4a). Some 3,280 biomarkers (40.6%) were significantly associated with monotherapy $IC_{50}$ including multiple previously described associations[5] such as taselisib (PI3K inhibitor) sensitivity in *PIK3CA*-mutant cell lines (Extended Data Fig. 4b). We identified 4,798 significant combination response biomarkers, of which 18.4%, 15.8% and 65.7% were associated with potency ($\Delta IC_{50}$), efficacy ($\Delta E_{max}$) and combination activity (combination $E_{max}$), respectively (Fig. 3a, Extended Data Fig. 4c). Of note, 76.8% of the $\Delta IC_{50}$ and $\Delta E_{max}$ biomarkers are associated with either $\Delta IC_{50}$ (43.0%) or $\Delta E_{max}$ (33.8%) but not with both, consistent with the complementary nature of these synergy metrics. Of the 2,025 combinations, 28.7% had at least one combination response biomarker (for combinations with one or more biomarkers: median: 2, range: 1–152; Extended Data Fig. 4d). We identified more than 2,050 combination biomarkers unique to the molecular basket analyses, demonstrating the benefit of testing for biomarkers within defined molecular subgroups.

Of the 1,645 $\Delta IC_{50}$ and $\Delta E_{max}$ associations, we identified only three in pancreas-specific analyses: of these, low *CDH1* gene expression,

which is associated with epithelial-to-mesenchymal transition, sensitised cells to irinotecan (TOP1 inhibitor) plus AZD7762 (CHEK1/2 inhibitor) (Extended Data Fig. 4e). Among known combination biomarkers, we identified *BRAF* mutation as a biomarker for dabrafenib (BRAF) paired with EGFR inhibitors afatinib and sapitinib screened in colon (Extended Data Fig. 4f), with synergy occurring exclusively in *BRAF*-mutant cell lines[16] (4 or 6 synergistic cell lines, respectively). Combinations of EGFR antibodies and BRAF inhibitors are a clinically approved regimen in *BRAF*-mutant metastatic colorectal cancer[17]. These examples show that our screen can identify known clinical biomarkers and candidate markers of combination response warranting further investigation.

To understand the relationship between drug targets and the molecular features in cells that influence drug synergy, we overlaid single-agent and combination biomarkers onto the previously described IntAct protein–protein interactome. We mapped 42.2% (633 out of 1,501) of features onto protein nodes; mapping was impossible for certain features such as clinical subtyping. We calculated the shortest finite network distance between each of the drug target and feature nodes for 582 and 124 unique combination and single-agent biomarkers, respectively (Extended Data Fig. 5a). Both types of biomarkers have a median shortest node distance of 3 (interactome diameter or maximum distance = 12); however, sensitivity biomarkers that were the drug target themselves (that is, a shortest distance of zero) were rarer for combinations (2.75%) than for single-agents (12.33%; Extended Data Fig. 5a). This was confirmed using Reactome[18], an alternative interactome (Extended Data Fig. 5b), and randomly shuffling biomarkers to simulate entirely false-positive associations eliminated the observation (Extended Data Fig. 5c for IntAct and Extended Data Fig. 5d for Reactome). These data are consistent with synergy being mediated through combined drug activity and indicate that drug combination biomarkers can be difficult to determine by single-agent activity alone.

Examples of combination biomarkers that are drug targets are high gene expression of *PARP1* in olaparib (PARP1/2 inhibitor) combinations, and a copy number gain of *CDK12*, *ERBB2* and *MED24* (feature: 'gain:cnaPANCAN301') as predictor of sensitivity to ERBB2-targeting combinations such as lapatinib (targeting EGFR and ERBB2) with vorinostat (a histone deacetylase inhibitor) in breast. Low expression of *NRAS* predicted response to dabrafenib (BRAF inhibitor) plus trametinib (MEK1/2 inhibitor) in breast, consistent with *NRAS* expression modulating BRAF inhibitor sensitivity[19]. This represents a biomarker one node away from a combination drug target (Fig. 3b). Mutation of *PIK3CA* predicted response to linsitinib (which targets IGF1R) + MK-2206 (AKT1/2 inhibitor) in a pan-tissue *KRAS*-mutant context, and is two nodes away from targets of both drugs (Fig. 3c).

Considering the combination–tissue pairs with more than three synergistic cell lines, we find that 164 out of 662 (24.8%) have at least one associated $\Delta E_{max}$ or $\Delta IC_{50}$ biomarker (Fig. 3d, Supplementary Table 4). We found several examples in which the identified feature associated with combination response is closely associated with synergy. For instance, all seven breast cell lines synergistic for the combination of sapitinib (EGFR and ERBB2/3 inhibitor) and JQ1 (an inhibitor of BRD2, BRD3, BRD4 and BRDT) show a gain in *ERBB2* (Fig. 3e). Similarly, six out of nine breast cell lines that show synergy for MK-2206 (AKT1/2 inhibitor) and alpelisib (PI3Kα inhibitor) have a mutation in *PTEN*. In a pan-tissue setting, *KRAS* mutation significantly associates with sensitivity to trametinib (MEK1/2 inhibitor) and MK-2206 (AKT1/2 inhibitor), and 74% of synergistic cell lines are *KRAS* mutants (Fig. 3f). These examples demonstrate that many synergistic combinations can be linked with a biomarker.

Our comprehensive analysis identified drug combination response biomarkers, including for many synergistic combinations, which could be used for patient stratification in pre-clinical and clinical follow-up. Combination biomarkers are more likely than single agents to be emergent properties arising from combinatorial drug activity in the context of signalling networks.

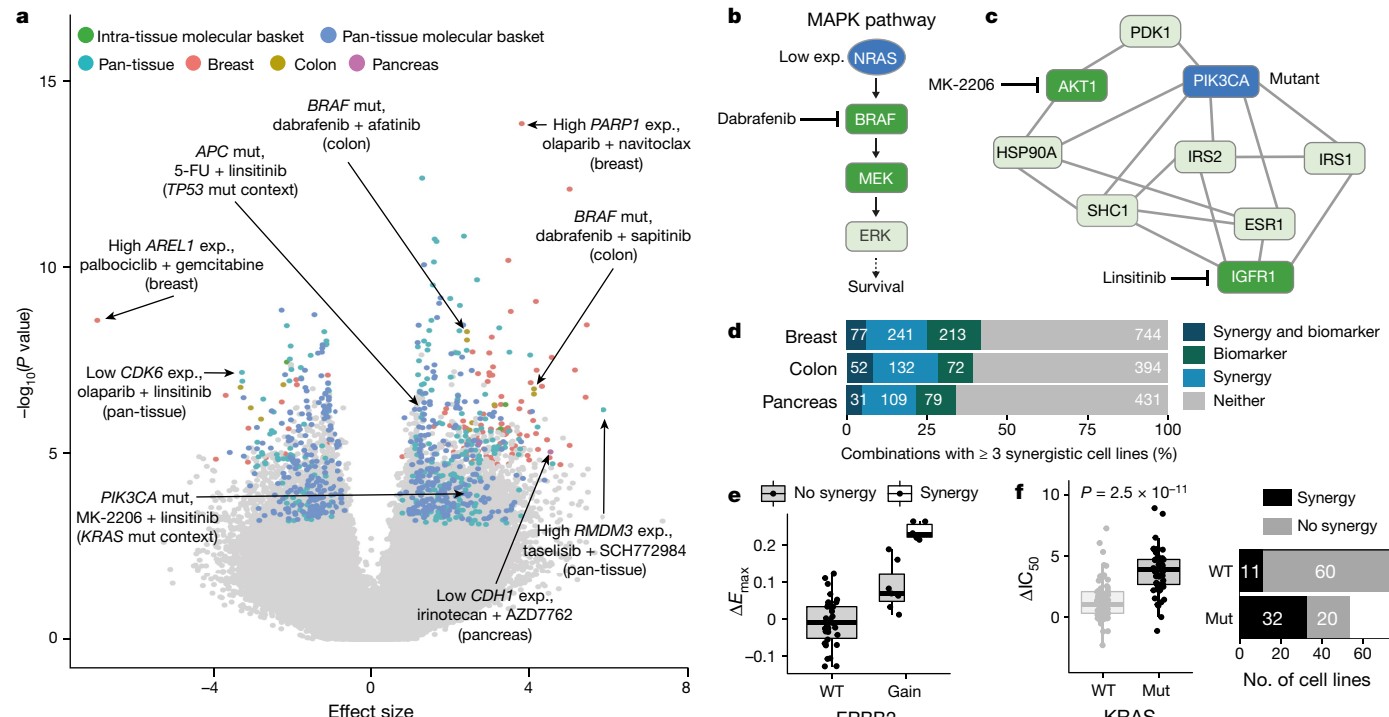

**Fig. 3 | A biomarker pipeline incorporating multi-omics features identifies context-specific associations. a**, Volcano plot of biomarkers associated with $\Delta IC_{50}$ ($n = 2,006,328$). Statistically significant large effect-size biomarkers ($n = 884$) are coloured by analysis type. Examples discussed in the text and selected outliers are labelled. Exp., expression; mut., mutant. **b**, Schematic of MAPK pathway showing the relationship between NRAS and BRAF. Low expression of NRAS is a biomarker of certain dabrafenib-containing combinations. **c**, Network of interactors of PIK3CA, showing its position two nodes away from targets of MK-2206 and linsitinib. *PIK3CA* mutation is predictive of the linsitinib + MK-2206 combination response in the

*KRAS*-mutant molecular context. **d**, Number of combinations with at least three synergistic cell lines and combination response biomarkers ($\Delta E_{max}$ or $\Delta IC_{50}$). **e**, Gain of *ERBB2* is associated with sapitinib + JQ1 combination response in breast and all synergistic cell lines have an *ERBB2* amplification. **f**, *KRAS* mutation is associated with trametinib + MK-2206 combination response in a pan-tissue setting (left, two-sided Welch's *t*-test) and most synergistic cell lines harbour mutated *KRAS* (right). In box plots, the horizontal line shows the median, boxes extend across first and third quartiles, and whiskers extend to 1.5× interquartile range.

## Combinations in cancers with unmet need

We leveraged the number and molecular diversity of the cell lines to investigate synergy rate, biomarkers and clinical trials for combinations screened in three populations with unmet clinical need: patients with basal-like breast cancer ($n = 22$), MSS colon cancer ($n = 31$) or *KRAS*-mutant colon cancer ($n = 25$).

We compared combination synergy rates within each of these populations with the other cell lines from the same tissue (Fig. 4a, Extended Data Fig. 6a, b). Between 3% and 5% ($n = 107$) of all combinations have synergy in at least 25% of cell lines from these populations (41 combinations for basal-like breast cancer, and 28 and 38 for MSS and *KRAS*-mutant colon cancer, respectively), with some having exquisite specificity of synergy in the population of interest. Of these 75 unique combinations, we identified 11 combinations with matching trials on clinicaltrials.gov for basal-like breast (10 trials) or MSS colon (1 trial) cancer (Fig. 4b, Supplementary Table 5). Cisplatin (a DNA crosslinker) combined with gemcitabine (a pyrimidine antimetabolite) or MK-1775 (WEE1 and PLK1 inhibitor; using MK-1775 as anchor) was highly synergistic in basal-like breast cancer (47% and 59% synergy rate, respectively) and both combinations are in clinical trials in clinically related triple-negative breast cancer (Fig. 4a). Synergy for cisplatin + gemcitabine was tested and robust in the validation screen (88% synergy overlap). Furthermore, several combinations showed repurposing potential: for instance, combined MK-1775 (WEE1 and PLK1 inhibitor) and irinotecan (TOP1 inhibitor) treatment, screened with camptothecin as the TOP1 inhibitor, had

a 26.7% synergy rate in MSS colon cancer (versus 6.7% in microsatellite unstable (MSI)), is currently in a trial in rhabdomyosarcoma and blastomas (NCT02095132). Phase 1 safety studies of navitoclax (BCL2, BCL-XL and BCL-W inhibitor) paired with chemotherapeutics such as gemcitabine, docetaxel and paclitaxel are ongoing, all of which had high synergy rates in basal-like breast (63%, 41%, and 38%; Fig 4a), with navitoclax + gemcitabine, the only one of the three that was part of the validation screen, having robust synergy (100% overlap between screens).

A third or more of top-ranking combinations in populations of unmet need had at least one $\Delta E_{max}$ or $\Delta IC_{50}$ biomarker, some of which were identified within the molecular basket that defines the population (Fig. 4b, Supplementary Table 5). For example, in the *KRAS*-mutant colon group, cell lines with loss of *ERCC3* were associated with sensitivity to linsitinib (IGF1R inhibitor) plus MK-2206 (AKT1/2 inhibitor) (Extended Data Fig. 6c). Our data can identify specific combinations in populations of unmet need, provide support for ongoing clinical trials, and identify biomarkers and repurposing opportunities for combinations already in trials.

## Irinotecan and CHEK1i synergize in colon

One of the top synergistic combinations was camptothecin (TOP1 inhibitor) with AZD7762 (CHEK1/2 inhibitor). Camptothecin is an analogue of the standard-of-care chemotherapeutic irinotecan used to treat colon cancer, and CHEK1 inhibitors can potentiate responses of DNA-damaging compounds through abrogation of DNA

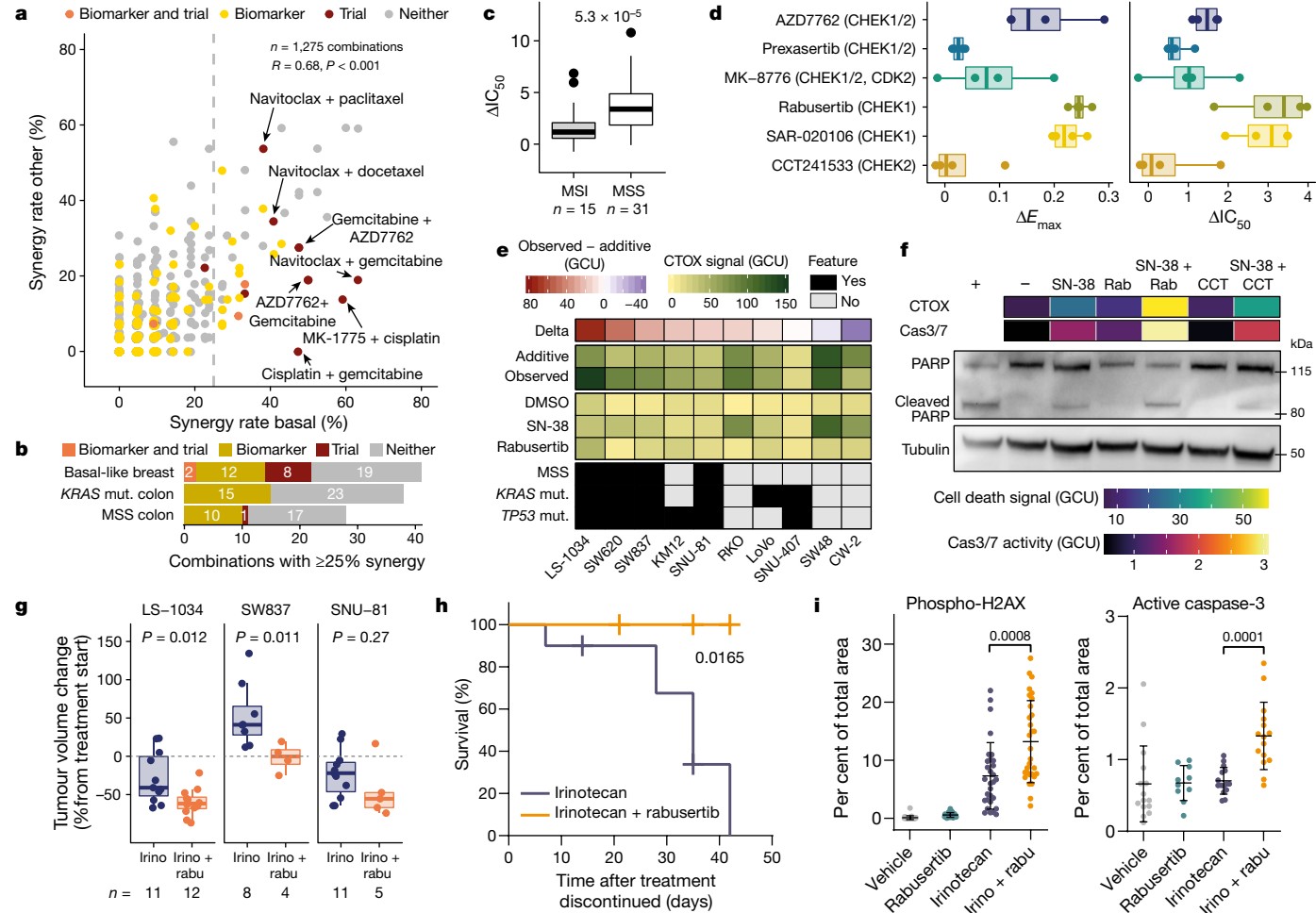

**Fig. 4 | Populations of unmet clinical need and validation of combined irinotecan and CHEK1 inhibitor treatment. a**, Synergy rates of combination treatments for breast cancer, comparing basal-like (*x*-axis) against other PAM50 subtypes (*y*-axis). Dashed line indicates a 25% synergy rate. *R*, Pearson correlation coefficient. Combinations with biomarkers or clinical trials are indicated. **b**, Combinations with at least 25% synergy with biomarkers or ongoing trials. **c**, AZD7762 (CHEK1/2 inhibitor) and camptothecin (TOP1 inhibitor) show higher potency ($\Delta IC_{50}$) in colon MSS cells. Replicates averaged and both combination anchor–library configurations pooled. Two-sided Welch's *t*-test. **d**, The response to combination treatment is CHEK1-specific. Activity of camptothecin combined with six CHEK inhibitors in 4 colon cell lines for 72 h. **e**, In most cases, combined TOP1 and CHEK1 inhibition produces cell death that is greater than the additive effect. CellTox green (CTOX) signal (in green calibrated units (GCU)) after 72 h of treatment with SN-38 (TOP1 inhibitor) and rabusertib (CHEK1 inhibitor). Mean of 3 or 4 biological replicates. Additive is defined as the sum of single-agent responses. Delta is observed response minus additive response. **f**, Inhibition of TOP1 and CHEK1 for 72 h

induces caspase-dependent cell death in SW837 cells. CTOX and caspase 3/7 (Cas3/7) activity is shown as the mean of 3 biological replicates. PARP western blot is representative of three independent experiments (+, positive control; −, DMSO-only negative control); rabu, rabusertib; CCT, CCT241533. **g**, Rabusertib increases irinotecan response in vivo in colon cancer cells engrafted in NOD/SCID mice. Two-tailed unpaired Welch's *t*-test. **h**, Addition of rabusertib to irinotecan treatment improves survival of mice. SNU-81 cells were engrafted in NOD/SCID mice; mice were treated with irinotecan (*n* = 10 mice) or irinotecan + rabusertib (*n* = 4) for 35 days and monitored for 42 days after treatment discontinued. Log-rank Mantel–Cox test; *P* value shown. **i**, Combined rabusertib and irinotecan treatment increases genotoxic stress. LS-1034 cells were treated as in **g**. Tumours were collected 72 h after start of treatment and stained for phospho-H2AX (*n* = 30) and active caspase 3 (rabusertib, *n* = 10; other groups, *n* = 15). Data are mean ± s.d. Two-tailed unpaired Welch's *t*-test. In box plots, the horizontal line shows the median, boxes extend across first and third quartiles, and whiskers extend to 1.5× interquartile range.

damage-induced cell cycle arrest[20–22]. This combination yielded high synergy rates in MSS colon cancer cell lines (62.1% and 53.3% for both orientations), with significantly higher potency and efficacy in MSS cell lines than in MSI cell lines (Student's *t*-test, *P* < 0.005; Fig. 4c, Extended Data Fig. 6d, Supplementary Table 5). Furthermore, *KRAS*-mutant colon cell lines showed high synergy rates for this combination (46% and 48%) and *KRAS–TP53* double-mutant cell lines had significantly stronger combination responses than *KRAS* single-mutant cell lines (Extended Data Fig. 6e, Supplementary Table 5). Thus, we identified two potential patient populations, MSS and *KRAS–TP53* double-mutant colon cells, showing notable benefit from inhibition of both CHEK1/2 (CHEKi) and TOP1 (TOP1i).

We next combined camptothecin with six CHEK1/2 inhibitors with different selectivity. CHEK1-selective inhibitors SAR-020106 and rabusertib produced large shifts in potency (median $\Delta IC_{50}$: 8.5- to 10.5-fold shift) and efficacy (median $\Delta E_{max}$: 0.22–0.24) in combination with camptothecin in 4 cell lines, whereas the CHEK2-selective inhibitor CCT-241533 did not (Fig. 4d). Combining SN-38, the active metabolite of the TOP1 inhibitor irinotecan, with small interfering RNA (siRNA) against CHEK1, but not CHEK2, resulted in a synergistic viability reduction (Extended Data Fig. 7a–d), and at least sevenfold reduction in the $IC_{50}$ of SN-38 (Extended Data Fig. 7e). Our results indicate that combination effects of TOP1i + CHEKi are primarily mediated through inhibition of CHEK1. This is corroborated by reports of the potentiating effect

of CHEK1 inhibition, but not CHEK2 inhibition, with DNA-damaging agents such as topoisomerase 1 inhibitors[23].

In colony formation assays, combining low concentrations of SN-38 with rabusertib (CHEK1 inhibitor) resulted in fewer colonies and increased cell death (Fig. 4e, Extended Data Fig. 7f–i) than either drug alone, or when combining SN-38 with CCT241533 (CHEK2). Cell death effects of SN-38 + rabusertib surpassed an additive response for many colon cell lines (Fig. 4e), particularly in cell lines with weak to moderate response to SN-38 alone, consistent with CHEK1 inhibition potentiating the effect of TOP1 inhibition. The combination effects ranged from less than additive to robust potentiation, with all MSS and most *KRAS*–*TP53* double-mutant cell lines showing at least additive response (Fig. 4e). The combination induced activation of the apoptosis markers caspase 3/7 and PARP cleavage (Fig. 4f, Extended Data Fig. 7g, i; For gel source data, see Supplementary Fig. 1).

We engrafted three colon cancer cell lines (LS-1034, SW837 and SNU-81) in NOD/SCID mice and treated them for 24–35 days with irinotecan (TOP1 inhibitor), rabusertib (CHEK1 inhibitor), or with a combination of the two drugs. In LS-1034 and SW837, which showed more cell death with combined TOP1i and CHEK1 inhibition than SNU-81 in vitro (Fig. 4e), the effect of the combination therapy on end-of-treatment tumour volumes and on tumour growth inhibition over time was more pronounced than that for irinotecan alone (Welch's *t*-test, $P < 0.05$) (Fig. 4g, Extended Data Fig. 8a). In SNU-81, although the response to combination therapy was more similar to treatment with irinotecan alone, the resumption of tumour growth after drug withdrawal was delayed in mice treated with the combination compared to those treated with irinotecan alone (log-rank Mantel–Cox test; end point 750 mm$^3$), suggesting a fitness disadvantage (Fig. 4h). Combination treatment led to more DNA double strand breaks (phospho-H2AX positive cells), less proliferative and more apoptotic tumour cells than in irinotecan-treated LS-1034 tumours 72 h after treatment start (Fig. 4i; Extended Data Fig. 8b).

These data validated combined TOP1 and CHEK1 inhibition as a potent combination in MSS and *KRAS*–*TP53* double-mutant colon cancer cells, and demonstrate the potential for follow-up of other synergistic drug combinations identified here.

## Discussion

The scale and breadth of our study provides insights into combination response. We establish that evaluating combination potency and efficacy are complementary, a recently introduced concept[10], and could identify combinations leading to dose reduction, improved efficacy or both, relative to single agents. We demonstrate that synergy is rare, as has been described[6,7], that it varies within tissues and between molecular backgrounds, and that combinations of targeted drugs are more likely to synergise than combinations involving chemotherapy. This indicates that combinations of targeted agents in molecularly defined patient populations are most likely to be synergistic. We identified many highly synergistic combinations, notably for cancers of unmet clinical need. We recommend detailed validation of promising drug combinations reported here. As proof of concept, we validated in vitro and in vivo a combined irinotecan and CHEK1 inhibition. Although combinations of CHEK1i with DNA-damaging agents have been linked to *TP53*- and *KRAS*-mutant cancer[24,25], to our knowledge, this is the first report of notable activity in MSS and *KRAS*–*TP53* double-mutant colon cancer. Clinical trials combining CHEKi with chemotherapy show variable toxicity and anti-tumour activity, particularly for unstratified patient populations. Since irinotecan is approved for the treatment of colon cancer, and rabusertib (a CHEK1-selective inhibitor) has an acceptable safety profile in phase 1 trials[26], our data indicate that this combination—with appropriate consideration of potential toxicity—could be particularly effective for patients with MSS or *KRAS*–*TP53* double-mutant colon cancer.

The data presented here are a rich resource and augment existing genomic, transcriptomic and functional datasets for cell lines available as part of the Cancer Dependency Map[27]. Future screens in additional cancer types, focused studies using more complex culture models, and screening of higher-order combinations, will support and extend our observations. The testing of combinations in non-cancer cell lines may help to estimate clinical toxicity. Our findings that drugs with weak single-agent activity, and those separated by one or two nodes in a protein–protein interaction network are most likely to yield a synergistic interaction could improve the design of future screens. Similarly, our data could improve computational approaches, which currently lack training datasets, for predicting effective drug combinations in different contexts[4,28,29]. In conclusion, the data and analyses presented here are fertile ground for catalysing new discoveries and a basis for effective rational combinatorial therapies.

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

## Methods

### Statistics

Statistical tests are two-sided Welch's $t$-tests unless otherwise specified. Enrichment analyses were performed using the phyper function for hypergeometric tests in R. Multiple testing correction was performed based on the Benjamini–Hochberg method. Biomarker analysis was performed using GDSCTools as described below. Box plots depict the median value as the centre bar, first and third quartiles as box boundaries, and whiskers extending to either first quartile minus $1.5 \times$ the inter-quartile range (lower boundary), or third quartile plus $1.5 \times$ the inter-quartile range (upper boundary). Points beyond this region are individually plotted.

### Cell lines

Cell lines were acquired from commercial cell banks. All cells were grown in RPMI medium (supplemented with 10% FBS, 1% penicillin/streptomycin, 1% glucose, 1 mM sodium pyruvate) or DMEM/F12 medium (supplemented with 10% FBS, 1% penicillin/streptomycin) (Supplementary Table 2) at 37 °C in a humidified atmosphere at 5% $CO_2$. To prevent cross-contamination or misidentification, all cell lines were profiled using a panel of 94 single nucleotide polymorphisms (Fluidigm, 96.96 Dynamic Array IFC). Short tandem repeat (STR) analysis was also performed, and cell line profiles were matched to those generated by the cell line repository. All cell lines are routinely tested for mycoplasma and are negative for mycoplasma. Further information on the cell lines used in this study, including their source and molecular profiling datasets can be found in Supplementary Table 2 and in the Cell Model Passports database[30] (https://cellmodelpassports.sanger.ac.uk).

### Compounds

Compounds were sourced from commercial vendors (Supplementary Table 1). DMSO-solubilized compounds were stored at room temperature in low humidity (<12% relative humidity), low oxygen (<2.5%) environment using storage pods (Roylan Developments). Water-solubilized compounds were maintained at 4 °C. For 8 compounds their identities and purity were confirmed by UHPLC-MS. Identity was confirmed by mass spectrometry (6550 iFunnel Q-TOF LC/MS, Agilent Technologies) using electrospray ionization in positive and/or negative modes. Anchor and library concentrations were drug- and tissue-specific and determined using a two-step process. First, drug concentrations were selected based on primary literature, in vitro data of minimum concentrations inhibiting relevant target activity and viability[31], clinical data indicating achievable human plasma concentrations, or where known concentrations that induce sensitivity in a biomarker positive cell line. Additionally, a pilot screen, testing a 1,000-fold concentration range of each drug in 9–13 cell lines per tissue (breast: 13, colon: 9, pancreas: 12), was performed and concentrations optimized to give a range of sensitivities across the cell lines. Anchor drugs were screened at two fixed concentrations with a 2-, 4- or 10-fold difference between them to give moderate activity (50–90% viability) across the cell lines within each cancer type. Screening concentrations (Supplementary Table 1) typically did not exceed 10 μM and were in the range of human plasma exposures achievable in patients[32]. Library drugs were screened at seven concentrations spanning a 1,000-fold range with a non-equidistant $\log_2$ design of four 4-fold steps followed by two 2-fold dilution steps starting at the lowest used concentration. The use of this design was based on the observation that higher concentrations were most informative and would benefit from denser profiling. As an alternative to μM concentration ranges, drug concentrations and $IC_{50}$ values can be visualized on a standardized $\log_2$ scale, with 9 being equivalent to the highest screened concentration. Each screening plate contained five replicates of the anchor alone (high and low concentrations) and four replicates of the library alone (full dose response). A single replicate of the combination dose response was performed in the primary screen.

### Screening

Cells were transferred into 1,536-well plates in 7.5 μl of their respective growth medium using XRD384 (FluidX) dispensers. The seeding density was optimised prior to screening to ensure that each cell line was in the exponential growth phase at the end of the assay. For this, six seeding densities with a two-fold dilution step were each dispensed into 224 wells of a single 1,536-well assay plate (XRD384 (FluidX) dispenser) and cells were incubated for 96 h. Cell number was quantified using CellTiter-Glo 2.0 (Promega). The maximum density tested varied based on cell type, typically 5,000 cells per well for suspension cells and 1,250 cells per well for adherent cells (Supplementary Table 1). Assay plates were incubated at 37 °C in a humidified atmosphere at 5% $CO_2$ for 24 h then dosed with the test compounds using an Echo555 (Labcyte). Final DMSO concentration was typically 0.2%. Following dosing with compounds assay plates were incubated, and the drug treatment duration was 72 h. To monitor cell growth over the duration of drug treatment, a parallel undrugged control plate was assayed at the time of drug treatment and referred to as a 'day = 1' plate. This was repeated each time that a cell line was screened. To measure cell viability, 2.5 μl of CellTiter-Glo 2.0 (Promega) was added to each well and incubated at room temperature for 10 min; quantification of luminescence was performed using a Paradigm (Molecular Devices) plate reader.

### Assay plate quality control

All screening plates contained negative control wells (untreated wells, $n = 6$; DMSO-treated wells, $n = 126$) and positive control wells (blanks—that is, medium-only wells, $n = 28$; staurosporine-treated wells, $n = 20$; and MG-132 treated wells, $n = 20$) distributed across the plate. We used these positive and negative control wells to test whether the plates meet defined quality control criteria. A maximum threshold of 0.18 was applied to the coefficient of variation (CV) of the DMSO-treated negative controls (CV $= \sigma_N/\mu_N$, where $\sigma_N$ is the s.d. of the negative control and $\mu_N$ is the mean of the negative control). Using the DMSO-treated negative control (NC1) and the two positive controls (PC1 and PC2), we determined $Z$-factors (also known as $Z'$; $Z$-factor $= 1 - 3 \times (\sigma_P + \sigma_N)/(|\mu_P - \mu_N|)$, where $\sigma_N$ and $\sigma_P$ are the s.d. of the negative and positive controls, respectively, and $\mu_N$ and $\mu_P$ are the mean of the negative and positive controls, respectively). The $Z$-factors were calculated for all plates that indicate sensitivity of the cell lines to the positive control (ratio of NC1:PC $\geq 4$). In case a cell line is insensitive to both positive control drugs, the $Z$-factors were calculated based on blank wells instead. $Z$-factors were required to exceed a minimum threshold of 0.3 for individual plates and a mean of 0.4 across all plates within a screening set. Where a cell line was sensitive to both positive controls, it had to pass $Z$-factor thresholds for both positive controls. Plates that did not meet these requirements were excluded from the study. Overall, 3,106 (>70%) of 1,536-well microtitre screening plates passed coefficient of variation and $Z$-factor thresholds. Wherever possible, failed plates were repeated, leading to dataset completeness of more than 96% for all three tissues (breast: 96.5%, colon: 99.8% and pancreas: 99%).

### Curve fitting

For each plate, the raw fluorescent intensity values were normalised to a relative viability scale (ranging from 0 to 1) using the blank ($B$) and negative control (NC) values (viability = (Fluorescence of treated cells $- B$)/(NC $- B$)). Anchor viability was determined from the mean across the five replicate wells screened on each plate. All library drug dose responses were fitted as a two-parameter sigmoid function[33]. The dose–response curves for the combinations were fitted similarly, but with two notable differences: (1) the cell line parameters were obtained from the library drug fits; (2) the maximum viability was capped at the anchor viability (rather than from 0 to 1). We use the 50% (inflection) point of the sigmoidal curve between zero and the anchor viability for both the expected Bliss and the observed combination.

We extended the model to nest each replicate within the drug or cell line to obtain stable estimates from the replicate experiments. To assess the quality of the fits, we computed the root mean square error (RMSE) and excluded curves with RMSE > 0.2 (equalling 1.5% of measurements). The $E_{max}$ and the $IC_{50}$ are based on the fitted curves. $E_{max}$ is reported at the highest tested concentration for the drug.

## Classifying synergy

To detect synergy we compared observed combination responses to expected combination responses. For the latter, we used Bliss independence[9] of the response to the anchor and the library drug alone. Conceptually, every point on the Bliss dose response curve is defined as the product between the anchor viability and the corresponding point on the library dose response curve. Shifts in potency ($\Delta IC_{50}$) and in efficacy ($\Delta E_{max}$) were calculated as the difference between the observed combination response and expected Bliss ($\Delta IC_{50}$ = Bliss $IC_{50}$ − combination $IC_{50}$, and $\Delta E_{max}$ = Bliss $E_{max}$ − combination $E_{max}$). $\Delta IC_{50}$ is reported on a $\log_2$ scale.

A given measurement was synergistic if the combination $IC_{50}$ was less than twice the highest screened library concentration and either the $\Delta IC_{50}$ or the $\Delta E_{max}$ was above a specific threshold: $\Delta IC_{50} \geq 3$ ($2^3$ is equivalent to an 8-fold shift in $IC_{50}$) or the $\Delta E_{max} \geq 0.2$ (20% shift in viability). Replicate measurements of 'anchor-library–cell line' tuples were summarized as synergistic if half or more of the replicate measurements showed synergy. To summarize both anchor concentrations, we considered a 'combination–cell line' pair as synergistic if synergy was observed at either anchor concentration.

## Reproducibility

To assess the reproducibility within a screen, we generated 2–18 biological replicates for 4–5 cell lines per tissue (breast: 5 (AU565, BT-474, CAL-85-1, HCC1937, MFM-223); colon: 4 (HCT-15, HT-29, SK-CO-1, SW620); pancreas: 5 (KP-1N, KP-4, MZ1-PC, PA-TU-8988T, SUIT-2)). Single-agent and combination responses were averaged across technical replicates (typically 3 per biological replicate) and correlated (Pearson correlation coefficient; minimum of 322 biological replicate pairs per 'metric-tissue' pair).

To assess the reproducibility of the screen, we rescreened a subset of combinations in each tissue (breast: 51 combinations in 34 cell lines; colon: 45 combinations in 37 cell lines; pancreas: 59 combinations in 29 cell lines; Supplementary Table 2). Drug combination responses were averaged across replicates within a screen and key metrics of single-agent and combination response were correlated between the two screens (Pearson correlation coefficient). To determine the quality of synergy calls, the original screen was considered as ground truth and numbers of true positive (TP), false positive (FP), true negative (TN) and false negative (FN) synergistic combination–cell line pairs were calculated. These were used to calculate $F$-score ($F$-score = TP/(TP + 0.5 × (FP + FN))), recall (recall = TP/(TP + FN)), and precision (precision = TP/(TP + FP)) per tissue. To investigate the strength of effects of $\Delta E_{max}$ and $\Delta IC_{50}$ of FP and FN measurements, the distance to $\Delta E_{max}$ and $\Delta IC_{50}$ synergy thresholds was calculated for each 'anchor concentration-library–cell line' tuple based on combination responses averaged across replicates ($n = 9,570$ tuples).

## Biomarker analysis

Matrices of single-agent (library $IC_{50}$) and combination response (combination $E_{max}$, $\Delta IC_{50}$, $\Delta E_{max}$) metrics were used as input for GDSCTools ANOVAs[15]. To obtain a single combination $E_{max}$, $\Delta IC_{50}$ and $\Delta E_{max}$ value per cell line–combination pair, responses were averaged across replicates for each anchor concentration-library–cell line tuple and the combination metrics were compared for the two anchor concentrations: the larger of the two $\Delta IC_{50}$ and $\Delta E_{max}$ values and the smaller of the two combination $E_{max}$ values were used for biomarker discovery in order to capture the largest effects of the combination. A range of binary feature

files were used, including multi-omics binary event matrices (MOBEMs) composed of genes known to be mutated, amplified or homozygously deleted in human cancers[5] (number of features = 1,073), CELLector signatures[34] ($n = 227$ for breast, $n = 261$ for colon), RNA-seq gene expression[35] ($n = 1,184$; original dataset accession number E-MTAB-3983), CRIS[36] and PAM50[37,38] classifications. Gene expression was limited to a curated panel of genes composed of targets of the drugs used, additional members of the *BCL2* family and apoptosis-associated genes[39], genes annotated as clinically relevant for cancer[40], and genes whose mutations were listed as features in the MOBEMs[5,40] and CELLector[5,34,40] feature files. Continuous values of gene expression were binarized by $z$-scoring each variable across the subset of cell lines used for the molecular context tested, and substituting a $z$-score $\geq 2$ for a binary value representing that feature being elevated (that is, 'Gene_up'), and a $z$-score $\leq -2$ for a binary feature representing that feature being decreased (that is, 'Gene_down'). Overall significance thresholds were $P \leq 0.001$ and FDR $\leq 5\%$.

## Network overlays

An interactome of binary, undirected interactions was built in the iGraph R package (https://cran.r-project.org/web/packages/igraph/citation.html) using the Reactome[18] human interactions file (accessed April 2021), and all human interactions reported in IntAct[12] (accessed July 2021). All non-protein nodes and duplicated interactions were removed, resulting in a non-directed network of 5,556 Uniprot protein nodes and 25,731 edges for the Reactome interactome. For the IntAct interactome, an evidence filter of 0.5 was applied, and all non-protein nodes and duplicate interactions were removed, resulting in a non-directed network of 14,431 protein nodes and 110,118 edges. Drug targets and biomarkers features were manually mapped to their Uniprot proteome identifiers (UPID), with overall 57 out of 66 (86.3%) drug target profiles being mapped to one or more UPIDs, and 633 out of 1,501 (42.2%) biomarker features being mapped for one or more UPIDs. UPID mapping was not possible for chemotherapeutics, PAM50, CRIS, and not done for methylation sites not associated with a cancer driver gene. A distance matrix between all nodes was calculated using iGraph: infinite values were reported for nodes that did not exist in the same network. When calculating the shortest distance between drug targets or drug targets and biomarkers, distances were calculated for all target-target or target-biomarker pairs and the smallest distance was reported. For example, for a drug with two targets combined with a drug with three targets, the shortest of six target-target distances would be reported. To simulate false positive biomarker associations, the biomarker features used in the genuine distance plot were randomly shuffled without replacement, before re-calculating the shortest distance between the new, 'false' biomarker and drug targets.

## Clinical trials

Clinical trials data were extracted from the API at https://clinicaltrials.gov/ (accessed March 2021) using an R script and searches in the format 'drug1 + drug2 + cancer + tissue'. Obtained lists of trials were manually curated to ensure that drugs were exact matches and to remove trials using radiotherapy alongside drug combination treatment. Searches were limited to 81 combinations with $\geq 25\%$ synergy in populations of clinical need.

## Specificity of CHEK inhibition

To test CHEK specificity we seeded SW620, SW837, SNU-81 or LS-1034 cells in 96-well plates (770–2,750 cells per well) and treated them with camptothecin (anchor, 0.025 μM) in combination with six CHEK inhibitors (libraries, all dosed at 1 μM highest used concentration unless indicated): AZD7762 (CHEK1, CHEK2), prexasertib (CHEK1, CHEK2), MK-8776 (CHEK1, CHEK2, CDK2), SAR-020106 (CHEK1), rabusertib (CHEK1) and CCT241533 (CHEK2; 2 μM). After 96 h of drug treatment

viability was measured with CellTiter-Glo 2.0 (CTG; Promega). Drug response curves were fitted as described above.

For siRNA experiments, SW837 and SNU-81 cells (8,000 and 16,000 cells per well, respectively) were reverse transfected with siRNAs of a non-targeting pool as negative control (siNT; Dharmacon, D-001810-10-05), polo-like kinase 1 (PLK1) pool as positive control (Dharmacon, L-003290-00-0010), CHEK1 pool (Dharmacon, L-003255-00-0005) or CHEK1 individual siRNAs (LQ-003255-00-0005), and CHEK2 pool (Dharmacon, L-003256-00-0005) using lipofectamine RNAiMax (Thermofisher). After 30 h, 0.025 μM or a dose range of 0.001–9 μM SN-38 or DMSO were added and viability was measured 72 h later with CTG. Signal was normalised to siNT+DMSO controls. Statistical significance between conditions was tested using a two-sided Welch's $t$-test.

## Real time cell death and caspase-3/7 activity

Cells were seeded in 96-well plates (typically 5,000–16,000 cells per well). After 24 h drugs (0.125 μM staurosporine (positive control), 0.025 μM SN-38, 0.75 μM rabusertib, 0.75 μM CCT241533) or DMSO and real-time fluorescent reagents for detection of cell death (CellTox Green; 1:1,000; Promega) or caspase-3/7 activity (IncuCyte Caspase-3/7 Red; 1:1,000; Essen Bioscience) were added. Pictures were recorded every 2 h for 96 h using an Incucyte (Essen Bioscience). Recorded fluorescent signals were measured as mean intensity per cell area and normalised to time 0 h.

## Colony formation

Cells were seeded in 6-well plates at 50,000 cells per well. Drugs (0.1 nM SN-38, 0.5 μM rabusertib, 0.5 μM CCT241533) or DMSO were added on day 1 and were refreshed through medium change on day 8. 14 days after drug treatment started the cells were fixed in 4% paraformaldehyde (Sigma-Aldrich) in PBS for 10 min at room temperature, and then stained with Giemsa (10%; Sigma-Aldrich) for at least 30 min at room temperature.

## Western blot

SW837 (1 million) or SNU-81 (1.5 million) cells were seeded in 10 cm dishes and treated with drugs (0.025 nM SN-38, 1.5 μM rabusertib, 1.5 μM CCT241533, 2 μM MG-132 (positive control)) or DMSO the day after. After 72 h, live and dead cells were collected and lysed in RIPA buffer (Sigma-Aldrich) supplemented with 1 mM DTT (Cayman Chemicals) and protease and phosphatase inhibitors (Roche). Total protein content was determined using Bradford reagent (ThermoFisher) and around 20 μg of lysate were loaded onto a 4–12% Bis-Tris gel (Invitrogen) for SDS–PAGE followed by protein transfer from the gel onto a PVDF membrane. Membranes were blocked in 5% milk (in TBST) and incubated overnight with the appropriate antibodies. Blots were washed in TBST and incubated with secondary antibody for 1 h at room temperature. Blots were washed in TBST before the signal was enhanced with Super Signal Dura and visualised. The following primary antibodies were used for immunoblot analysis: anti-PARP (Cell Signaling Technologies, 9542, 1:1,000; rabbit), and anti-β-tubulin (Sigma-Aldrich, T4026, 1:5,000; mouse) as loading control.

For experiments with knockdown of *CHEK1* and *CHEK2*, SW837 or SNU-81 cells were reverse transfected with siNT, siCHEK1 or siCHEK2 as described above. Cells were collected 72 h after transfection and lysed in RIPA buffer (Sigma-Aldrich, R0278) and protein concentrations were determined using the BCA assay (Novagen, 71285-3) as per manufacturer's instructions. SDS–PAGE and western blots were conducted as described above and the following primary antibodies were used for immunoblot analysis: anti-CHEK1 (Santa Cruz Biotechnology, sc-8408, 1:200; mouse), anti-CHEK2 (Cell Signaling Technologies, D9C6, 1:1,000; rabbit), and anti-β-actin (Abcam, ab6276, 1:5,000; mouse) as a loading control. Anti-Mouse IgG (GE Healthcare, NA931) and anti-rabbit (GE Healthcare, NA934; 1:2,000) HRP-linked secondary antibodies were used as secondary antibodies. PageRuler Plus Prestained Protein

Ladder, 10–250 kDa (ThermoFisher, 26620) was used as a molecular weight marker.

## In vivo tumour xenograft studies

A total of $4.5 × 10^6$ LS-1034 cells, $5 × 10^6$ SW837 cells or $2.5 × 10^6$ SNU-81 cells in 30% Matrigel were injected subcutaneously into the right flank of male and female 6-week-old NOD/SCID mice. Once tumours reached an average volume of approximately 300–400 mm³, mice were randomized into treatment arms, with $n = 12$ (irinotecan and irinotecan + rabusertib) or $n = 6$ (vehicle and rabusertib) per group. Rabusertib was administered orally, 200 mg kg⁻¹ daily (vehicle: 16.66% Captisol; CyDex, in 25 mM phosphate buffer, pH 4); irinotecan was administered intraperitoneally, 25 mg kg⁻¹ twice a week (vehicle: physiological saline). Tumour size was evaluated once weekly by calliper measurements, and the approximate volume of the mass was calculated using the formula $(4\pi/3) × (d/2)^2 × (D/2)$, where $d$ is the minor tumour axis and $D$ is the major tumour axis. Results were considered interpretable when a minimum of 4 mice per treatment group reached the prespecified endpoints (at least 3 weeks on therapy or development of tumours with average volumes larger than 2,000 mm³ within each treatment group in trials aimed to assess drug efficacy; at least 3 weeks after treatment cessation or development of individual tumours with volumes larger than 750 mm³ in survival experiments aimed to assess tumour control by therapy). A major tumour axis of 20 mm is the endpoint permitted by the Italian Ministry of Health in authorization no. 806/2016-PR, in accordance with national guidelines and regulations. This endpoint was not exceeded in any experiment. Operators were blinded during measurements. In vivo procedures and related biobanking data were managed using the Laboratory Assistant Suite[41]. All animal procedures were approved by the Institutional Animal Care and Use Committee of the Candiolo Cancer Institute and by the Italian Ministry of Health.

Statistical significance for tumour volume changes during treatment was calculated using a two-way ANOVA. For endpoint comparisons, statistical analysis was performed by two-tailed unpaired Welch's $t$-test. Statistical analyses in the survival experiments were performed by log-rank (Mantel–Cox) test. For all tests, the level of statistical significance was set at $P < 0.05$. Graphs were generated and statistical analyses were performed using the GraphPad Prism (v9.0) statistical package.

## Immunohistochemistry

Morphometric quantitation of Ki67, active caspase-3, and phospho-H2AX immunoreactivity was performed in xenografts from mice treated with vehicle (until tumours reached an average volume of 1,500 mm³) or the indicated compounds (after 72 h). Tumours ($n = 1–3$ for each treatment arm) were explanted and subjected to histological quality check and immunohistochemical analysis with the following antibodies: mouse anti-Ki-67(MIB-1) (Dako, GA626, 1:100), rabbit anti-cleaved caspase-3 (Asp175) (Cell Signaling, 9661, 1:200) and rabbit anti-phospho-histone H2AX (Ser139) (20E3) (Cell Signaling, 9718, 1:400). After incubation with secondary antibodies, immunoreactivities were revealed by DAB chromogen (Dako). Images were captured with the Leica LAS EZ software using a Leica DM LB microscope. Morphometric quantitation was performed by ImageJ software using spectral image segmentation. Software outputs were manually verified by visual inspection of digital images. Each dot represents the value measured in one optical field (40× for Ki67 and phospho-H2AX; 20× for active caspase-3), with 2–10 optical fields (Ki67 and phospho-H2AX) and 3–5 optical fields (active caspase-3) per tumour depending on the extent of section area ($n = 12–30$ for Ki67 and phospho-H2AX; $n = 8–15$ for active caspase-3). The plots show mean ± s.d. Statistical analysis by two-tailed unpaired Welch's $t$-test.

## Reporting summary

Further information on research design is available in the Nature Research Reporting Summary linked to this paper.

## Data availability

All drug sensitivity data generated for this study are included in this published article (and its supplementary information files) or available in a Figshare repository (https://doi.org/10.6084/m9.figshare.16895371, https://doi.org/10.6084/m9.figshare.16843600 and https://doi.org/10.6084/m9.figshare.16843597) and GDSC Combinations database (https://gdsc-combinations.depmap.sanger.ac.uk/). The cell line genomic datasets are available from the Cell Model Passports database (http://cellmodelpassports.sanger.ac.uk/). The following databases were accessed for this study: IntAct database (http://www.ebi.ac.uk/intact), the Reactome database (https://reactome.org/), the Cell Model Passports database (http://cellmodelpassports.sanger.ac.uk/) and the GDSC database (http://www.cancerrxgene.org/). Users have a non-exclusive, non-transferable right to use data files for internal proprietary research and educational purposes, including target, biomarker and drug discovery. Excluded from this licence are use of the data (in whole or any significant part) for resale either alone or in combination with additional data/product offerings, or for provision of commercial services. Source data are provided with this paper.

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

**Acknowledgements** We thank M. Michaut and R. Rahman for their work on CombiXplorer, a Shiny app initially used to explore combination drug response curves; G. Bounova for her CMS subtyping of colorectal cancer cell lines. Part of Fig. 1a was created using BioRender.com. We thank J. Mettetal, C. Crafter, A. Cheraghchi-Bashi and K. Bulusu for feedback on a precursor to the GDSC Combination data visualisation website, and U. McDermott for contributing to the original design of screens. This research was funded in whole, or in part, by Wellcome Trust Grant 206194. For the purpose of open access, the author has applied a CC BY public copyright licence to any Author Accepted Manuscript version arising from this submission. A. Bertotti and L.T. are supported by AIRC, Associazione Italiana per la Ricerca sul Cancro, Investigator Grants 20697 (to A. Bertotti) and 22802 (to L.T.); AIRC 5x1000 grant 21091 ; AIRC/CRUK/FC AECC Accelerator Award 22795 (to L.T.); European Research Council Consolidator Grant 724748—BEAT (to A. Bertotti); H2020 grant agreement no. 754923 COLOSSUS (to L.T.); H2020 INFRAIA grant agreement no. 731105 EDIReX (to A. Bertotti); and Fondazione Piemontese per la Ricerca sul Cancro-ONLUS, 5x1000 Ministero della Salute 2016 (to L.T.).

**Author contributions** M.J.G., S.B., P.J., D.J.V., C.H.B. and N.A. contributed to the screening design. A. Beck, C.H., C.T., E.L., F.T., I.M., J.H., J.M., L.R., S.B. and T.M. contributed to screening data collection. H.L., D.J.V. and D.v.d.M. contributed to the data processing. S.B., H.L. and P.J. contributed to the quality control of the data. P.J. and E.A.C. contributed to the data analysis of drug interactions. E.A.C. and P.J. contributed to the biomarker analysis. P.J., O.E. and E.F.C. conducted siRNA experiments and colony formation and apoptosis assays. L.D. conducted siRNA experiments including western blots. S.M.L., F.S., A. Bertotti and L.T. conducted the in vivo follow-up experiments, including immunohistochemical stainings. D.v.d.M., W.Y., H.L., E.A.C. and P.J. designed and implemented the drug combination website. P.J., E.A.C., D.J.V., L.W. and M.J.G. wrote the manuscript. All authors read and approved the manuscript.

**Competing interests** M.J.G. has received research grants from AstraZeneca, GlaxoSmithKline, and Astex Pharmaceuticals, and is founder of Mosaic Therapeutics. C.H.B is an employee of Novartis and previously received research funding from Novartis. L.T. reports research grants from Symphogen, Servier, Pfizer, Menarini, Merck KGaA and Merus. Drug combinations described in this study are subject to patents filed by Genome Research Limited, which is the name under which the Sanger Institute operates.

**Additional information**
**Correspondence and requests for materials** should be addressed to Mathew J. Garnett.

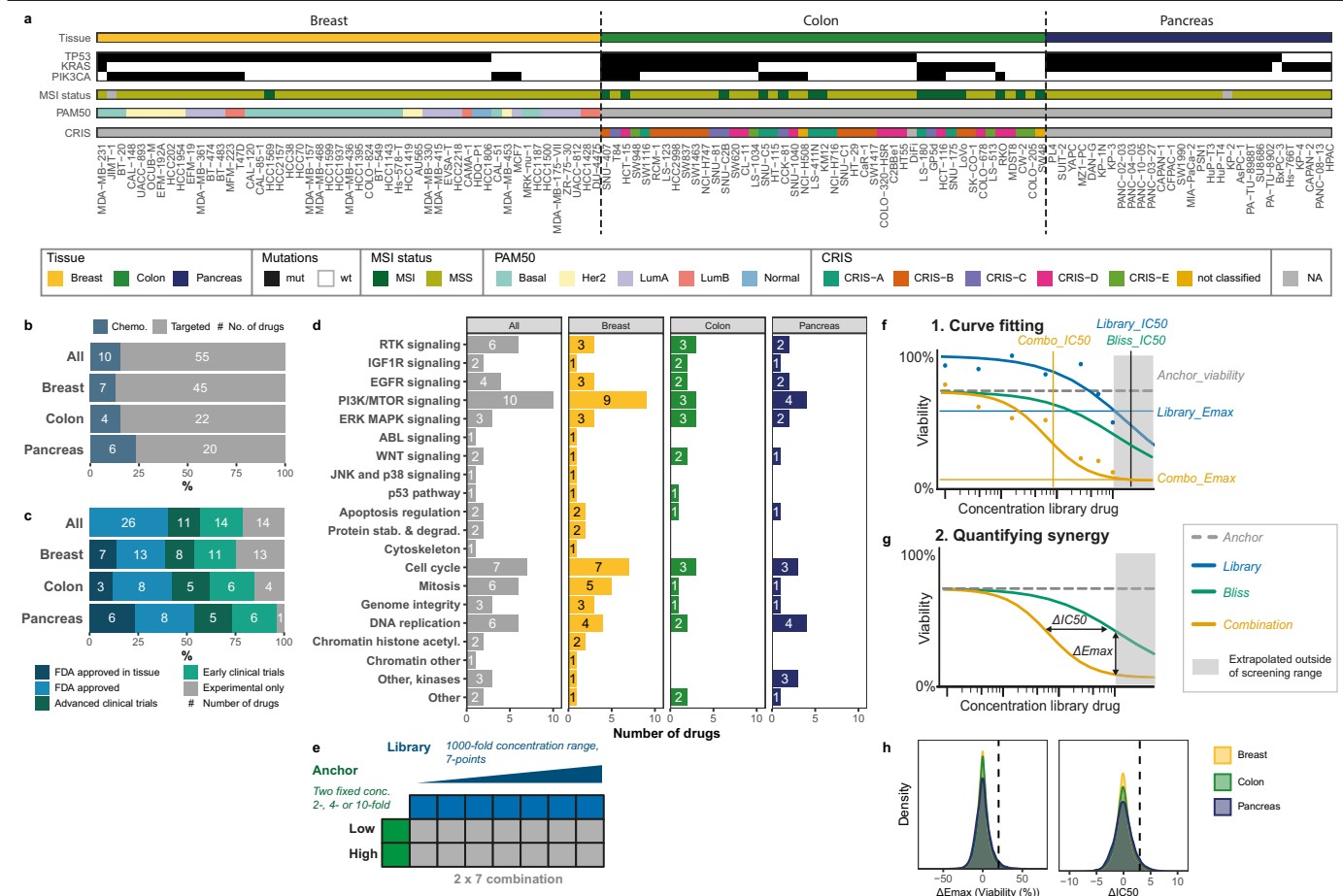

**Extended Data Fig. 1 | Information on cell lines, drugs and screen design.**
**a**, OncoPrint detailing mutation status of three key mutations (*TP53*, *KRAS*, *PIK3CA*), MSI status, and clinical subtyping (PAM50 or CRIS where available) for all 125 cancer cell lines. **b**, Proportion of chemotherapeutic and targeted drugs screened. **c**, Proportion of drugs that are FDA-approved, in clinical trials, or are in development. **d**, Number of drugs screened per pathway and tissue. **e**, Schematic of anchored screening design. An anchor is tested at two fixed concentrations against a library screened at a 7-point, discontinuous 1,000-fold concentration range (two 2-fold dilution steps from the highest used concentration, all other dilution steps are 4-fold). **f**, Schematic of drug

response curve fits of single-agent and combination responses. Vertical and horizontal lines are helper lines facilitating the reading of drug response metrics from the x-axis (concentration) and y-axis (viability), respectively. **g**, Schematic of synergy quantification based on efficacy (ΔEmax) or potency (ΔIC50). **h**, ΔEmax and ΔIC50 are normally distributed and only a minority meet synergy thresholds. Density distribution of ΔEmax (viability in %; left) and ΔIC50 (log2; right) across all combination responses. Vertical dashed lines represent synergy thresholds (ΔEmax ≥ 20% and ΔIC50 ≥ 3). n = 156,065 measurements in breast, n = 74,525 in colon, n = 66,117 in pancreas.

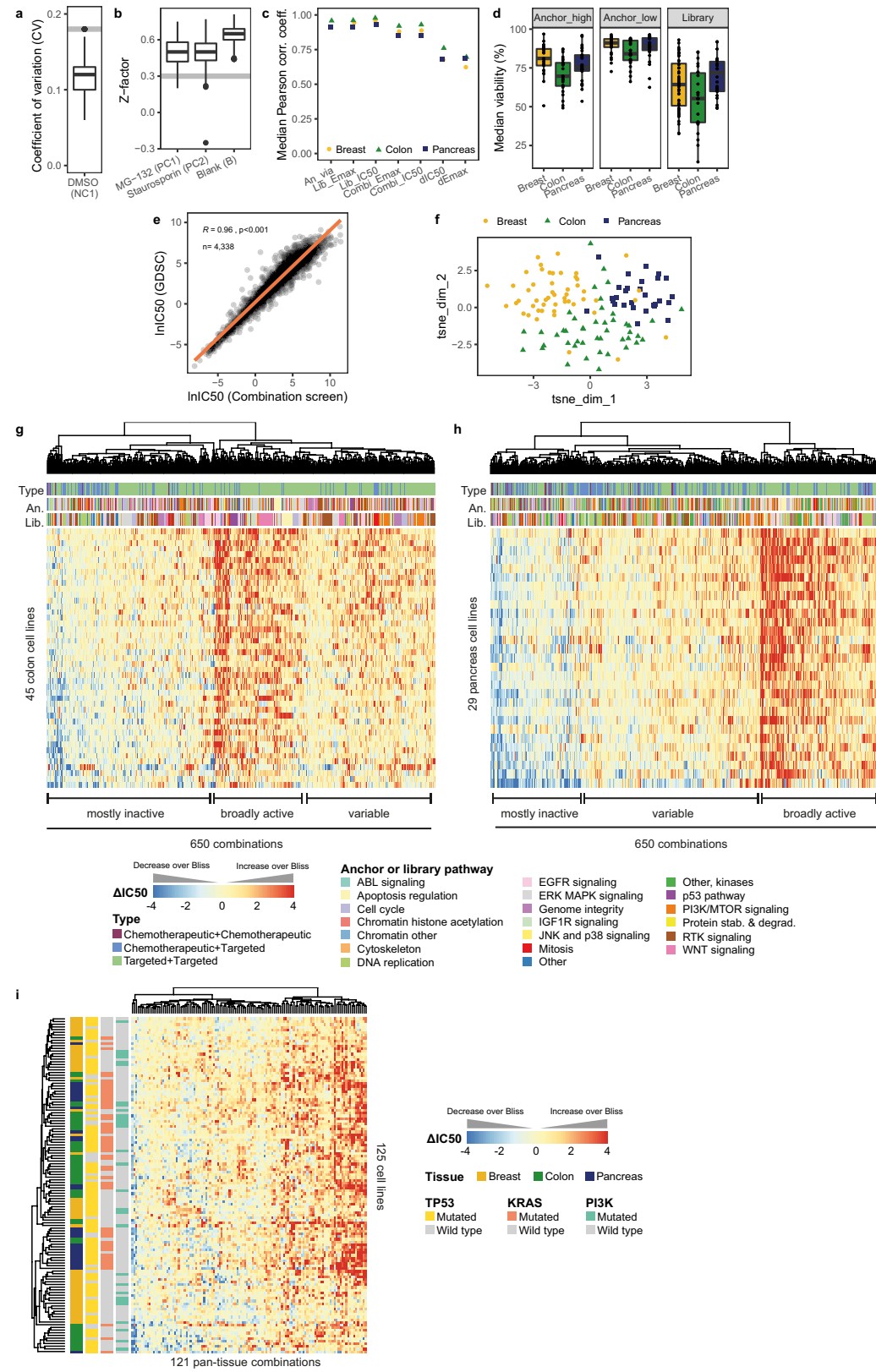

**Extended Data Fig. 2 |** See next page for caption.

**Extended Data Fig. 2 | Screen quality control, single-agent response and combination response trends. a**, The coefficient of variation of the negative control DMSO across all 3,106 drug screening plates is low. Grey dashed line represents the quality control threshold of CV < 0.18. Median and interquartile range (IQR). **b**, The plate Z-factor scores for positive control drugs (MG-132, staurosporine) and blank wells across all drug screening plates. Grey dashed line represents the plate threshold of Z-factor > 0.3. Median and IQR. **c**, Single-agent and combination responses are well correlated across biological replicates (r > 0.6, p-value < 0.05). Replicate data was collected over the duration of screening at various time points for 4-5 cell lines per tissue. Drug responses were averaged across technical replicates and correlated across biological replicates (n = 2–18, median = 4 biological replicates per 'anchor concentration-library-cell line' tuple) using Pearson correlation coefficient with Fisher's Z transform as statistical test. **d**, Monotherapies captured an informative range of drug response at the concentration selected, generally having weak to moderate activity in cell lines. Median anchor (anchor viability by anchor concentration) and library (library Emax) responses across cell lines within a tissue. n = 26 drugs in colon and pancreas, n = 25 anchors and n = 52 libraries in breast. Median and IQR. **e**, Library IC50s were highly correlated with corresponding drug responses from the Genomics of Drug Sensitivity in Cancer, with IC50 on natural log scale. Pearson correlation coefficient, n = 4,338 drug-cell line pairs. **f**, Tissue accounts for some, but not all, variance in combination response. Dimensional reduction using t-SNE analysis on combination responses (ΔIC50) of 121 pan-tissue combinations across 125 cancer cell lines. **g**, **h**, Heatmaps of combination responses (ΔIC50) in 45 colon (**g**) or 29 pancreas (**h**) cell lines. Drug responses were clustered by combination and are annotated by combination type and anchor (An.) and library (Lib.) pathway. Rows are sorted by conditional mean ΔIC50 on cell line identity. n = 650 combinations. **i**, Heatmap of combination responses (ΔIC50) for 121 pan-tissue combinations across 125 breast, colon and pancreas cancer cell lines, clustered by combination and cell line. For all heatmaps ΔIC50 limits were clipped to −4 and 4.

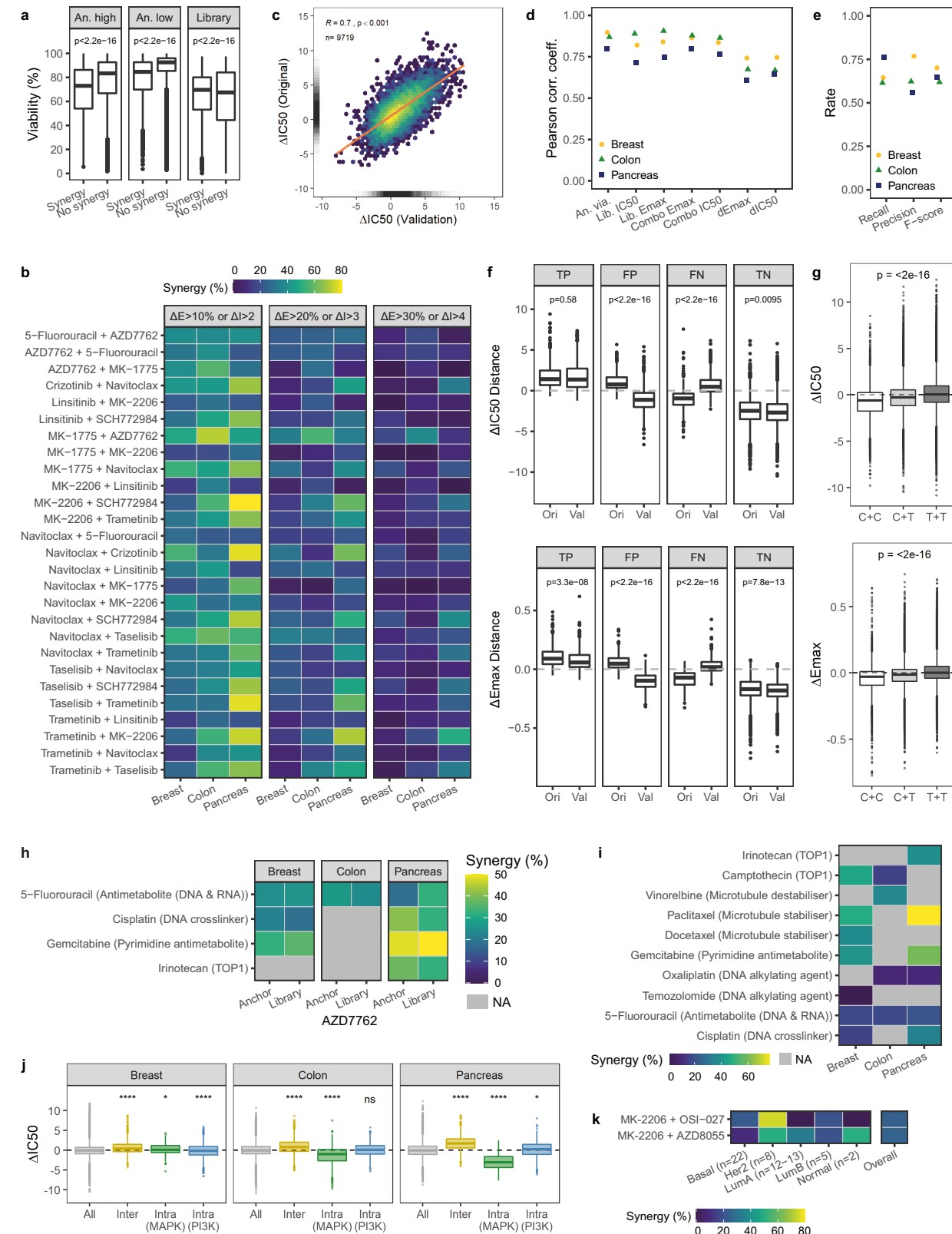

**Extended Data Fig. 3 | See next page for caption.**

**Extended Data Fig. 3 | Landscape of synergy. a**, Synergy is associated with weak to moderate single-agent drug activity. Single-agent activity (anchor low and high, library) of synergistic (n = 9,402) and non-synergistic (n = 287,305) measurements. Single-agent activity of synergistic measurements: Anchor high: 52 – 86% interquartile range (IQR), anchor low: 69 – 92% IQR, library: 53 – 80% IQR. Median and IQR. Two-sided Welch's t-test. **b**, The relative rate of synergy for the 27 pan-tissue combinations with > 20% synergy in at least one tissue remains context-dependent upon variation of synergy thresholds. Three different synergy thresholds of ΔEmax (ΔE) and ΔIC50 (ΔI) were applied. Thresholds used in this study were, at either anchor concentration, combination IC50 or Emax was reduced 8-fold or 20% viability over Bliss, respectively (central panel), and here compared to less (left) or more (right) stringent thresholds. **c**, ΔIC50 correlates well between the original and the validation screen. Drug responses were averaged across replicates within a screen. n = 9,719 'library-anchor concentration-cell line' tuples. Orange line represents linear fit. Pearson correlation coefficient and p-value. **d**, Single-agent and combination responses are well correlated between the original and validation screens. Drug responses were averaged across technical and biological replicates within a screen and correlated across screens. n > 3,000 'anchor concentration-library-cell line' tuples. **e**, Synergy classification is consistent. The F-score as well as the recall and precision rates were calculated for validated responses in breast (yellow; n = 1,651), colon (green; n = 1,597) and pancreas (blue; n = 1,633). **f**, False positives (FP) and false negative (FN) synergistic measurements have borderline ΔIC50 and ΔEmax values close to the threshold for calling synergy (compared to true positive (TP) and true negative (TN)). Drug responses were averaged across replicates for each 'anchor concentration-library-cell line' tuple. Distance to the synergy threshold (log 2 normalised ΔIC50 ≥ 3 or ΔEmax ≥ 0.2) was determined for each synergistic measurement. Median and interquartile range (IQR). Two-sided Welch's t-test. **g**, Combinations of two chemotherapeutics have lower combination responses. ΔIC50 and ΔEmax for chemotherapeutic+chemotherapeutic (C+C; n = 50), chemotherapeutic+targeted (C+T; n = 581) and targeted+targeted (T+T; n = 1,394) combinations. Median and IQR. ANOVA. **h**, AZD7762 has high synergy rates paired with certain chemotherapeutics. Synergy rate per tissue for AZD7762 paired with five chemotherapeutics in both anchor orientations. **i**, Several navitoclax+chemotherapeutic combinations have high synergy rates. Synergy rate per tissue for all combinations of navitoclax (anchor) paired with a chemotherapeutic (library). **j**, Inter-pathway targeting of MAPK and PI3K signalling leads to increased synergy effects. ΔIC50 per tissue for all combinations (All), inter-pathway MAPK and PI3K combinations (Inter), and intra-pathway combinations (Intra). Median and IQR. Two-sided Welch's t-test, *: p-value ≤ 0.05, **: p-value ≤ 0.01, ***: p-value ≤ 0.001, ****= p < 0.0001. **k**, Combinations of MK-2206 (AKT1, AKT2) and MTOR inhibitors are highly synergistic in Her2 breast cancer cell lines. Synergy rate of MK-2206 (anchor) paired with OSI-027 or AZD8055 (libraries) across all breast cancer cell lines and PAM50 subtypes.

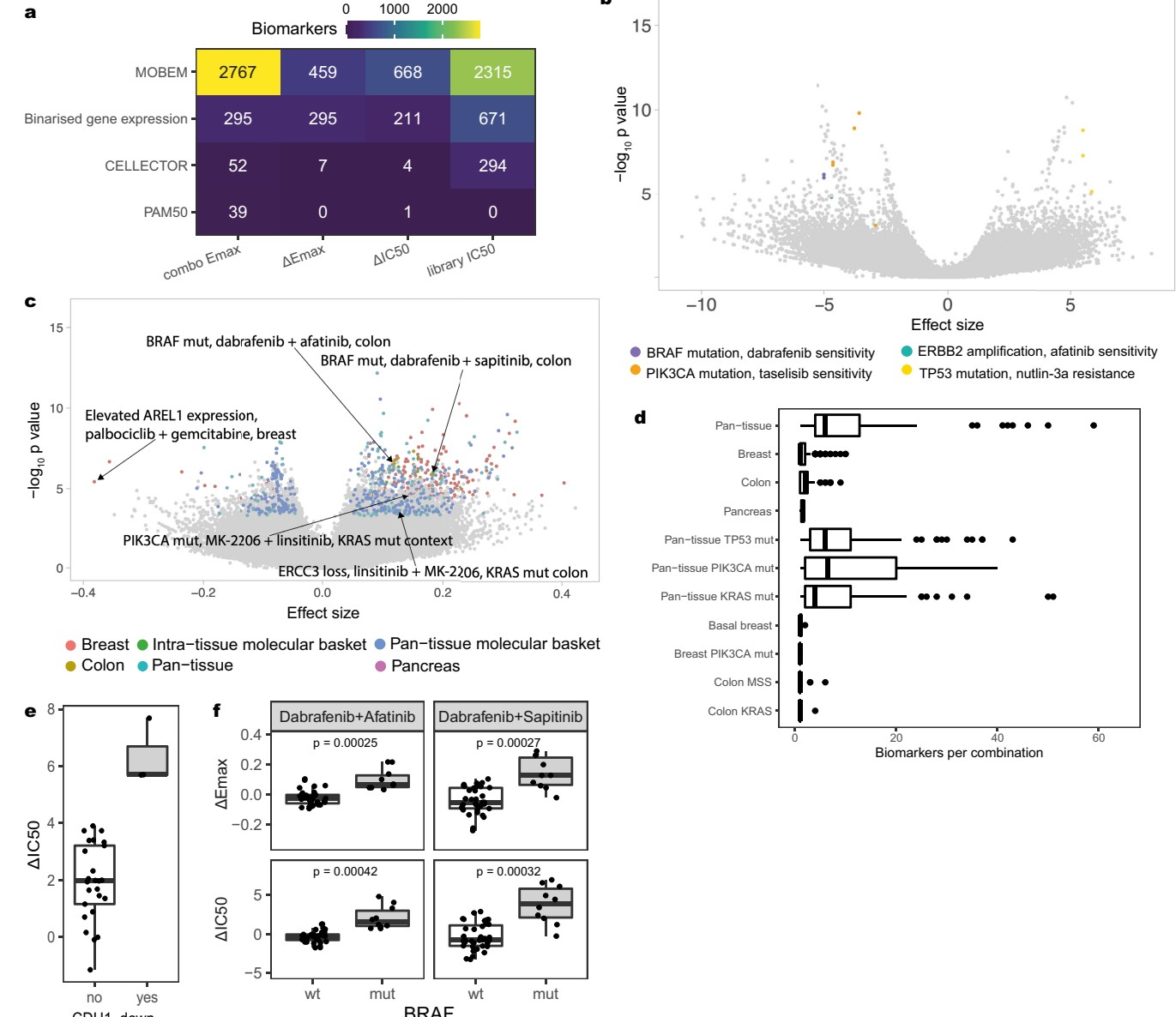

**Extended Data Fig. 4 | Landscape of biomarkers I. a**, Heatmap showing the distribution of 8,078 significant biomarkers across four inputs and four feature types. MOBEM: binary matrix of mutations, copy number alterations and methylations present in cell lines (see Methods for feature selection). **b**, Volcano plot of single-agent response biomarker associations tested (library IC50; n = 1,922,552; significant and large-effect biomarkers n = 3,280), with select statistically significant large-effect size biomarkers showing known single agent examples highlighted, namely *BRAF* mutation and dabrafenib sensitivity (purple), *ERBB2* amplification and afatinib sensitivity (turquoise), *PIK3CA* mutation and taselisib sensitivity (orange), and *TP53* mutation and resistance to nutlin-3a (yellow). Biomarkers identified using ANOVA test, p ≤ 0.001, FDR ≤ 5%, Glass deltas for positive and negative populations both ≥ 1. **c**, Volcano plot of biomarkers tested for associations with ΔEmax

(n = 2,006,328), with significant and large-effect biomarkers (n = 761) coloured by analysis type. Examples discussed in the text and selected outliers are labelled. Biomarkers identified using ANOVA test, p ≤ 0.001, FDR ≤ 5%, Glass deltas for positive and negative populations both ≥ 1. **d**, Number of significant and large-effect biomarkers found per combination per context. Median and interquartile range (IQR). **e**, ΔIC50 drug combination response for irinotecan+AZD7762 in pancreatic cell lines with low expression of CDH1. n = 3 CDH1_down, n = 27 not CDH1_down. ANOVA, p ≤ 0.001, FDR ≤ 5%. Median and IQR. **f**, Drug combination responses for dabrafenib paired with EGFR inhibitors afatinib or sapitinib in *BRAF* wild-type (wt; n = 36) and mutant (mut, n = 11) colon cell lines. ΔEmax and ΔIC50 were averaged across replicates and highest response between anchor concentrations is reported. Two-sided Welch's t-test. Median and IQR.

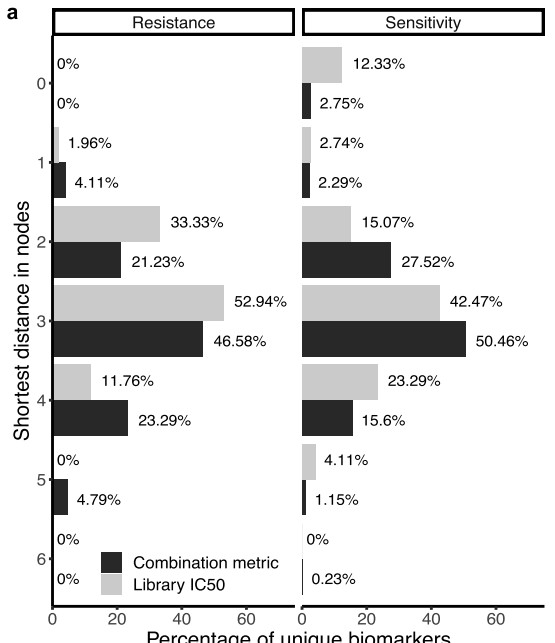

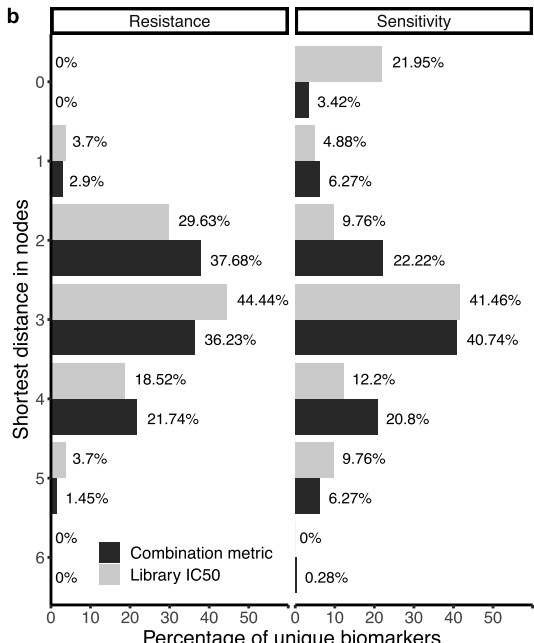

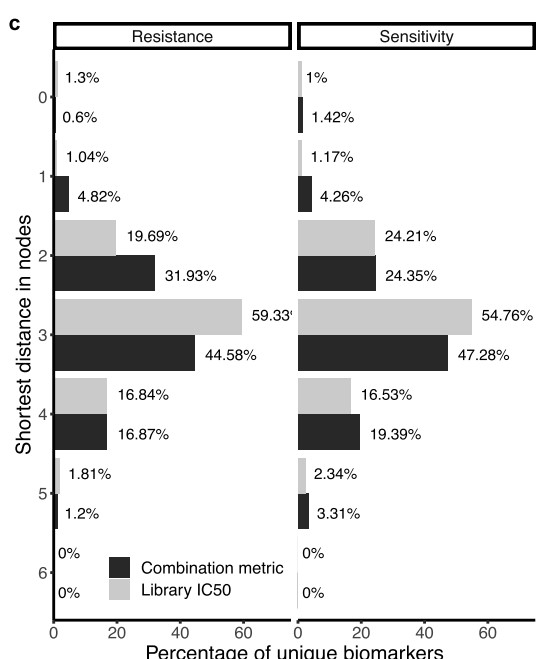

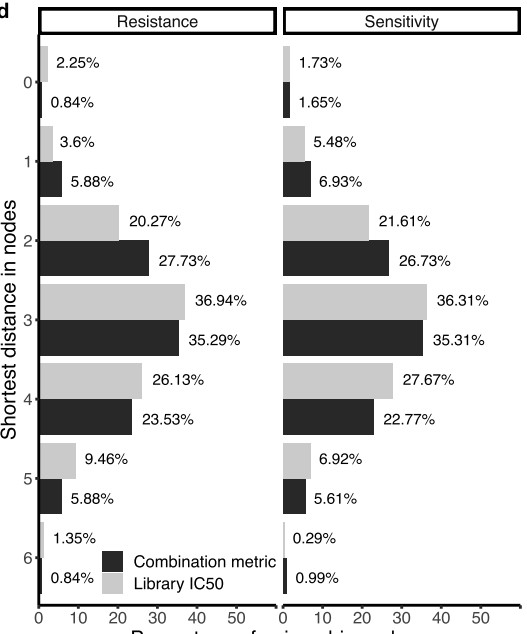

**Extended Data Fig. 5 | Landscape of biomarkers II. a**, Shortest distance in IntAct interactome between unique drug targets and biomarker features for combination metrics (ΔIC50 and ΔEmax biomarkers, n = 582) and single-agent library IC50 biomarkers (n = 124), split by whether the biomarker is associated with sensitivity or resistance. **b**, Shortest distance in Reactome interactome between unique drug targets and biomarker features for combination metrics (ΔIC50 and ΔEmax biomarkers, n = 420) and single-agent library IC50 biomarkers (n = 68), split by whether the biomarker is associated with sensitivity or resistance. **c**, Shortest distance in IntAct interactome between randomly shuffled unique drug targets and biomarker features for combination metrics (ΔIC50 and ΔEmax biomarkers, n = 589) and single-agent library IC50 biomarkers (n = 985), split by biomarker effect. **d**, Shortest distance in Reactome interactome between randomly shuffled unique drug targets and biomarker features for combination metrics (ΔIC50 and ΔEmax biomarkers, n = 422) and single-agent library IC50 biomarkers (n = 569), split by biomarker effect.

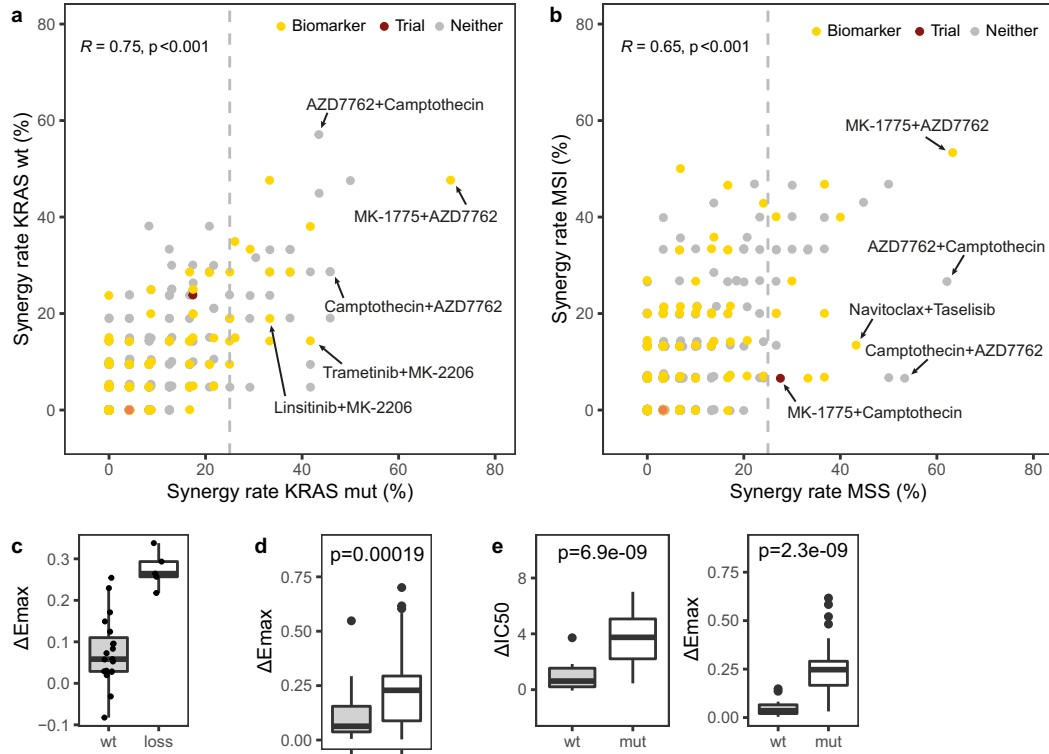

**Extended Data Fig. 6 | Populations of unmet clinical need. a**, **b**, 28 and 38
combinations are highly synergistic in *KRAS* mut (**a**) or MSS (**b**) colon cancer cell
lines and some have ΔEmax or ΔIC50 biomarkers or are in clinical trials. Synergy
rates of all colon combinations shown by *KRAS* mutant (mut; x-axis) versus
wild-type (wt; y-axis;). **b**) or MSS (x-axis) versus MSI (y-axis; **a**). Colours represent
biomarker or clinical trial presence. Vertical dashed line represents a synergy
rate of 25% in MSS or *KRAS* mutant cell lines. n = 650 combinations. Pearson
correlation with Fisher's Z transform as statistical test. **c**, Loss of *ERCC3* is
associated with increased efficacy (ΔEmax) of linsitinib+MK-2206 in *KRAS*
mutant colon (n = 20 wt; n = 5 loss). Median and interquartile range.
p-value < 0.001, FDR < 0.05. **d**, Colon MSS cells show higher efficacy (ΔEmax) for

AZD7762 (CHEK1/2) and camptothecin (TOP1). Drug combination responses
were averaged across replicates and both anchor-library combination
configurations were pooled (n = 31 MSS cell lines; n = 15 MSI cell lines). Median
and interquartile range. Two-sided Welch's t-test. **e**, AZD7762 and camptothecin
have greater potency (ΔIC50) and efficacy (ΔEmax) in *KRAS-TP53* double mutant
colon cancer cells (n = 8 *KRAS* mutant & *TP53* wild-type cell lines (wt); n = 16
*KRAS-TP53* double mutant cell lines (mut)). Drug combination responses were
averaged across replicates and both combination configurations were pooled.
Median and interquartile range. Two-sided Welch's t-test.

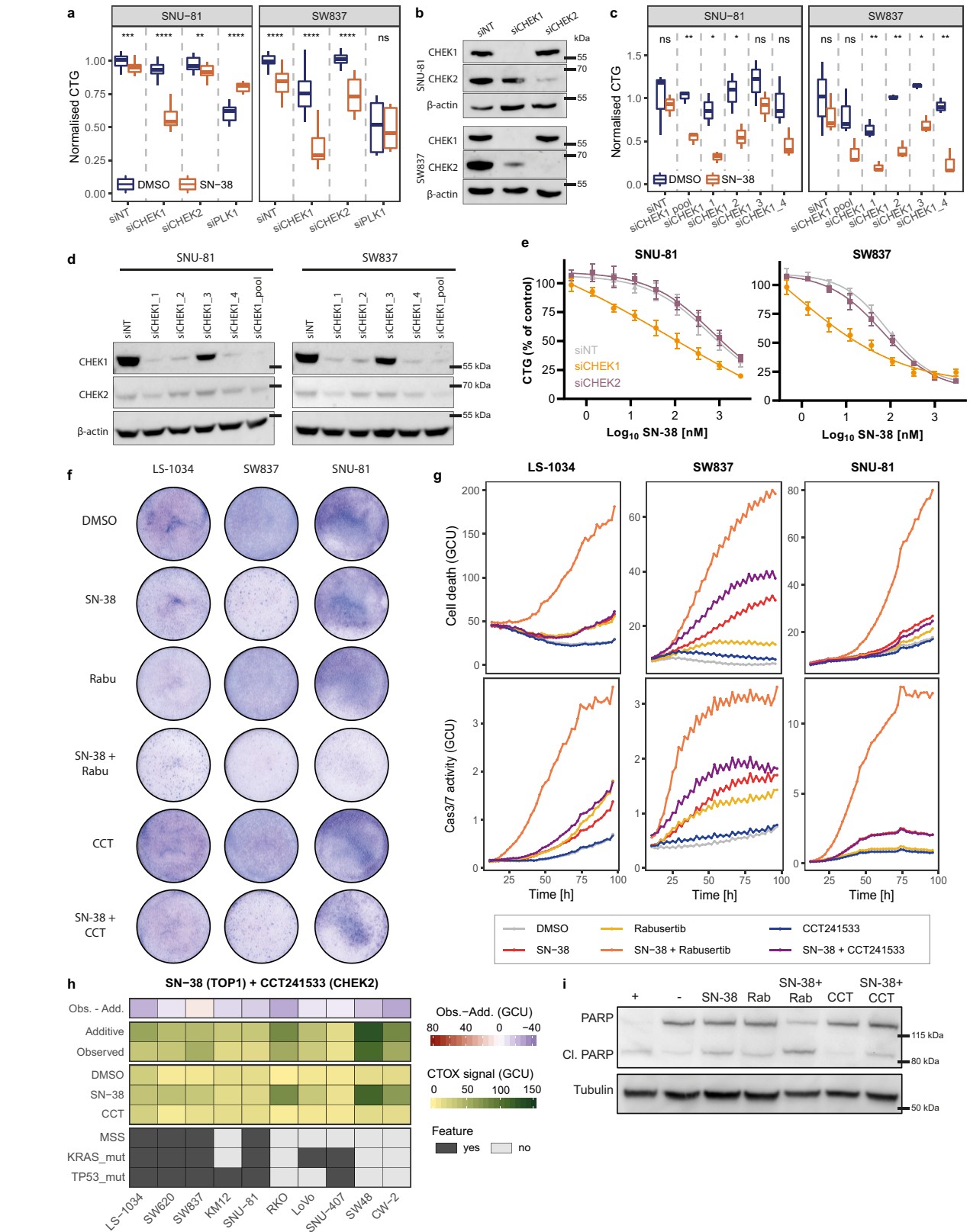

**Extended Data Fig. 7** | See next page for caption.

**Extended Data Fig. 7 | In vitro validation of combined targeting of TOP1 and CHEK1 in colon cancer. a**, Combination response is mostly CHEK1 specific. SW837 and SNU-81 cells were reverse transfected with pooled siRNA against CHEK1, CHEK2 or PLK1 (positive control), and 0.025 μM SN-38 was added 30 h later. Viability was measured with CellTiter-Glo (CTG) after 72 h of drug treatment. Signal was normalised to non-targeting siRNA (siNT)+DMSO controls. Median and interquartile range. Two-sided Welch's t-test, *= p ≤ 0.05, **= p ≤ 0.01, ***= p ≤ 0.001, ****= p ≤ 0.0001. **b**, CHEK1 and CHEK2 knockdown confirmation by Western blotting. SW837 and SNU-81 cells were reverse transfected with siRNAs (40 nM) and knockdown was examined after 72 h. Western blot is a representative of two independent experiments. For gel source data, see Supplementary Fig. 1. Some knockdown of CHEK2 was observed in SW837 cells with CHEK1 siRNA pool. **c**, CHEK1 specificity of combination response is confirmed with individual siRNAs against CHEK1. SW837 and SNU-81 cells were reverse transfected with pooled or four individual siRNA against CHEK1 and 0.025 μM SN-38 were added 30 h later. Viability was measured with CellTiter-Glo after 72 h of drug treatment. Signal was normalised to siNT+DMSO controls. Median and interquartile range. Two-sided Welch's t-test, *= p ≤ 0.05, **= p ≤ 0.01, ***= p ≤ 0.001, ****= p ≤ 0.0001. **d**, CHEK1 knockdown confirmation by Western blotting. SW837 and SNU-81 cells were reverse transfected with pooled or four individual siRNAs (40 nM) against CHEK1 and knockdown was examined after 72 h. Western blot is a representative of two independent experiments. For gel source data, see Supplementary Fig. 1. **e**, CHEK1 (but not CHEK2) silencing by siRNA significantly shifts and reduces the IC50 of SN-38 (SNU-81: 7.2-fold (IC50 siNT: 611.1 nM; siCHEK1: 84.4 nM (p = 0.0013); siCHEK2: 714.7 nM (p = 0.339)); SW837: 120-fold (IC50 siNT: 84.4 nM; siCHEK1: 0.69 nM (p = 0.0019); siCHEK2: 66.6 nM (p = 0.091)). SW837 and SNU-81 cells were reverse transfected with siRNAs and the following day cells were treated with a dose range of SN-38 (0.001–9 μM). Viability was assessed after 72 h using CellTiter-Glo. Signal was normalised to siNT+DMSO controls. Mean ± SD. Two-way ANOVA. **f**, Combination of rabusertib (CHEK1) and SN-38 reduces colony formation. Colon cancer cells were seeded and treated with drugs (0.1 nM SN-38, 0.5 μM rabusertib, 0.5 μM CCT241533) or DMSO for 14 days. CCT241533 is a CHEK2 selective inhibitor. Representative pictures of three experiments. **g**, Combination of rabusertib and SN-38 leads to caspase-mediated cell death. Colon cancer cells were seeded and treated with drugs (0.125 μM staurosporine (positive control), 0.025 μM SN-38, 0.75 μM rabusertib, 0.75 μM CCT241533) or DMSO in the presence of fluorescent reagents (CellTox-Green for cell death and IncuCyte Caspase-3/7 Red for caspase activity). Pictures were taken every 2 h for 96 h on the IncuCyte and fluorescent signals were measured as mean intensity per area and normalised to time 0 h. Mean of three independent experiments. **h**, Combined TOP1 and CHEK2 inhibition leads to mostly less than additive combination response. Cell death was measured by CellTox-Green signal (CTOX; in green calibrated units (GCU)) after 72 h of treatment with SN-38 (TOP1; 0.025 μM) and CCT241533 (CHEK2; 0.75 μM). Drug responses are mean across 3-4 biological replicates. Additive response: sum of SN-38 and rabusertib responses. Delta: observed - additive response. **i**, Combined TOP1 and CHEK1 inhibition results in PARP cleavage in SNU-81 cells. SNU-81 cells were treated with drugs for 96 h. Western blot is a representative of three repeated experiments. +: positive control (MG-132; 2 μM); -: negative control (DMSO; 1:1,000); SN-38 (0.025 μM); Rab: rabusertib (1.5 μM); CCT: CCT241533 (1.5 μM).

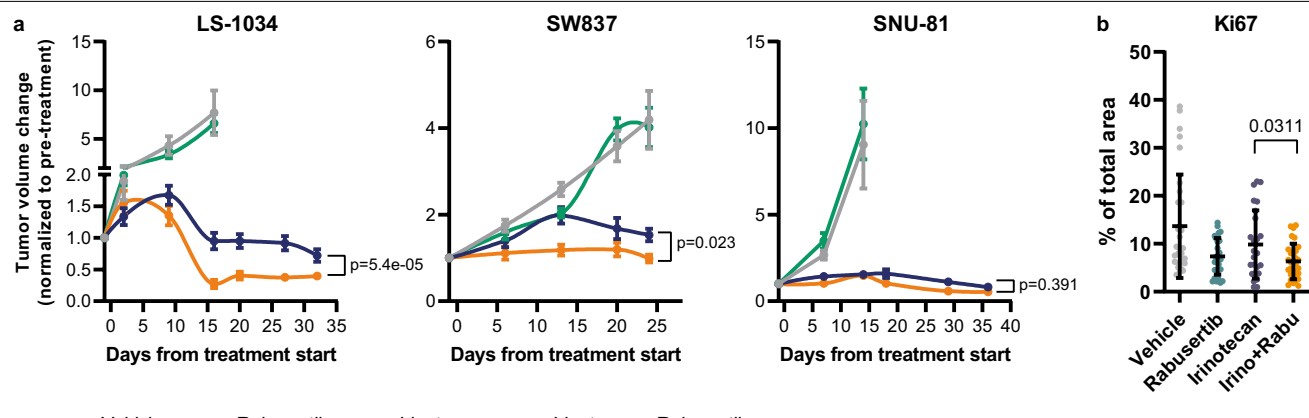

**Extended Data Fig. 8 | In vivo validation of combined targeting of irinotecan and CHEK1 in colon cancer. a**, Addition of rabusertib increases irinotecan response in two of three colon cancer xenograft models. NOD/scid mice were engrafted with colon cancer cell lines and treated with irinotecan (25 mg/kg twice a week) +/- rabusertib (200 mg/kg daily) for 24–35 days. Shown is average tumour volume change under treatment. LS-1034: n = 6 mice for vehicle, n = 11 for irinotecan, n = 12 for rabusertib and irinotecan+rabusertib. SW837: n = 6 mice for vehicle and rabusertib, n = 8 for irinotecan, n = 4 for irinotecan+rabusertib. SNU-81: n = 5 mice for vehicle and irinotecan+rabusertib, n = 6 for rabusertib, n = 10 for irinotecan. Two-way ANOVA. **b**, Treatment of rabusertib with irinotecan decreases proliferation. LS-1034 cells were engrafted and treated as described in (**a**). Tumours were collected 72 h after treatment start and stained for Ki67 (proliferative cells). n = 30 for vehicle, rabusertib and irinotecan+rabusertib; n = 25 for irinotecan. Mean ± SD. Two-tailed unpaired Welch's t-test.

# Reporting Summary

## Statistics

For all statistical analyses, confirm that the following items are present in the figure legend, table legend, main text, or Methods section.

| n/a | Confirmed | |
|---|---|---|
| ☐ | ☒ | The exact sample size (*n*) for each experimental group/condition, given as a discrete number and unit of measurement |
| ☐ | ☒ | A statement on whether measurements were taken from distinct samples or whether the same sample was measured repeatedly |
| ☐ | ☒ | The statistical test(s) used AND whether they are one- or two-sided<br>*Only common tests should be described solely by name; describe more complex techniques in the Methods section.* |
| ☐ | ☒ | A description of all covariates tested |
| ☐ | ☒ | A description of any assumptions or corrections, such as tests of normality and adjustment for multiple comparisons |
| ☐ | ☒ | A full description of the statistical parameters including central tendency (e.g. means) or other basic estimates (e.g. regression coefficient) AND variation (e.g. standard deviation) or associated estimates of uncertainty (e.g. confidence intervals) |
| ☐ | ☒ | For null hypothesis testing, the test statistic (e.g. *F*, *t*, *r*) with confidence intervals, effect sizes, degrees of freedom and *P* value noted<br>*Give P values as exact values whenever suitable.* |
| ☒ | ☐ | For Bayesian analysis, information on the choice of priors and Markov chain Monte Carlo settings |
| ☒ | ☐ | For hierarchical and complex designs, identification of the appropriate level for tests and full reporting of outcomes |
| ☐ | ☒ | Estimates of effect sizes (e.g. Cohen's *d*, Pearson's *r*), indicating how they were calculated |

*Our web collection on statistics for biologists contains articles on many of the points above.*

## Software and code

Policy information about availability of computer code

| Data collection | no software was used |
|---|---|
| Data analysis | The software package GDSC Tools v1.0.1 was used for biomarker analysis (https://doi.org/10.1093/bioinformatics/btx744). Dose response curves were fitted using a 2-parameter sigmoid function (https://doi.org/10.2217/pgs.16.15) |

For manuscripts utilizing custom algorithms or software that are central to the research but not yet described in published literature, software must be made available to editors and reviewers. We strongly encourage code deposition in a community repository (e.g. GitHub). See the Nature Portfolio guidelines for submitting code & software for further information.

## Data

Policy information about availability of data

All manuscripts must include a data availability statement. This statement should provide the following information, where applicable:

- Accession codes, unique identifiers, or web links for publicly available datasets
- A description of any restrictions on data availability
- For clinical datasets or third party data, please ensure that the statement adheres to our policy

All drug sensitivity data generated or analysed during this study are included in this published article (and its supplementary information files) or available in Figshare repository and GDSC Combinations database (https://gdsc-combinations.depmap.sanger.ac.uk/). Cell line metadata and genomic datasets are available from the Cell Model Passports database (https://doi.org/10.1093/nar/gky872). Users have a non-exclusive, non-transferable right to use data files for internal proprietary research and educational purposes, including target, biomarker and drug discovery. Excluded from this licence are use of the data (in whole or any significant part) for resale either alone or in combination with additional data/product offerings, or for provision of commercial services.

# Field-specific reporting

Please select the one below that is the best fit for your research. If you are not sure, read the appropriate sections before making your selection.

☒ Life sciences  ☐ Behavioural & social sciences  ☐ Ecological, evolutionary & environmental sciences

For a reference copy of the document with all sections, see nature.com/documents/nr-reporting-summary-flat.pdf

# Life sciences study design

All studies must disclose on these points even when the disclosure is negative.

| Sample size | For drug sensitivity testing and biomarker analyses we used all available cancer cell lines available. |
|---|---|
| Data exclusions | There were no data exclusions |
| Replication | To assess the reproducibility within a screen, we generated 2-18 biological replicates for 4-5 cell lines per tissue (breast: 5 (AU565, BT-474, CAL-85-1, HCC1937, MFM-223); colon: 4 (HCT-15, HT-29, SK-CO-1, SW620); pancreas: 5 (KP-1N, KP-4, MZ1-PC, PA-TU-8988T, SUIT-2)). Single-agent and combination responses were averaged across technical replicates (typically three per biological replicate) and correlated (Pearson correlation coefficient; minimum of 322 biological replicate pairs per 'metric-tissue' pair).<br><br>To assess the reproducibility of the screen, we rescreened a subset of combinations in each tissue (breast: 51 combos in 34 cell lines; colon: 45 combos in 37 cell lines; pancreas: 59 combos in 29 cell lines; Supplementary table 2). Drug combination responses were averaged across replicates within a screen and key metrics of single-agent and combination response were correlated between the two screens (Pearson correlation coefficient). To determine the quality of synergy calls, the original screen was considered as ground truth and numbers of true positive (TP), false positive (FP), true negative (TN) and false negative (FN) synergistic combination-cell line pairs were calculated. These were used to calculate F-score (F-score =TP / (TP + 0.5*(FP + FN))), recall (recall = TP / (TP + FN)), and precision (precision = TP / (TP + FP)) per tissue. To investigate the strength of effects of $\Delta$Emax and $\Delta$IC50 of FP and FN measurements, the distance to $\Delta$Emax and $\Delta$IC50 synergy thresholds was calculated for each 'anchor concentration-library-cell line' tuple based on combination responses averaged across replicates (n=9,570 tuples). |
| Randomization | For drug sensitivity testing and biomarker analysis, cell lines were organized into groups based on their their tissue and the presence of specific driver mutations or gene expression signatures. |
| Blinding | Blinding was not required as we used an unsupervised approach for biomarker analysis within defined cohorts. |

# Reporting for specific materials, systems and methods

We require information from authors about some types of materials, experimental systems and methods used in many studies. Here, indicate whether each material, system or method listed is relevant to your study. If you are not sure if a list item applies to your research, read the appropriate section before selecting a response.

## Materials & experimental systems

| n/a | Involved in the study |
|---|---|
| ☐ | ☒ Antibodies |
| ☐ | ☒ Eukaryotic cell lines |
| ☒ | ☐ Palaeontology and archaeology |
| ☐ | ☒ Animals and other organisms |
| ☒ | ☐ Human research participants |
| ☒ | ☐ Clinical data |
| ☒ | ☐ Dual use research of concern |

## Methods

| n/a | Involved in the study |
|---|---|
| ☒ | ☐ ChIP-seq |
| ☒ | ☐ Flow cytometry |
| ☒ | ☐ MRI-based neuroimaging |

# Antibodies

| Antibodies used | The following primary antibodies were used for immunoblot analysis: anti-PARP (Cell Signalling Technologies, 9542, 1:1,000; rabbit), anti-CHEK1 (Santa Cruz Biotechnology, sc-8408, 1:200; mouse), anti-CHEK2 (Cell Signaling Technologies, D9C6, 1:1000; rabbit), anti-β-tubulin (Sigma-Aldrich, T4026, 1:5,000; mouse) as loading control. Anti-Mouse IgG (GE Healthcare, #NA931) and anti-rabbit (GE Healthcare, #NA934) HRP-linked secondary antibodies were used as secondary antibodies. For immunohistochemical analysis with the following antibodies: anti-Ki-67(MIB-1)(Dako #GA626, 1:100; mouse), anti-cleaved caspase-3 (Asp175)(Cell Signaling #9661, 1:200; rabbit) and anti-phospho-histone H2AX (Ser139)(20E3)(Cell Signaling #9718, 1:400; rabbit). |
|---|---|
| Validation | All antibodies were validated by commercial vendors for Western blotting or immunohistochemistry, as required. anti-CHEK1 and anti-CHEK2 antibodies were validated using siRNA. |

# Eukaryotic cell lines

Policy information about cell lines

| | |
|---|---|
| Cell line source(s) | Cell lines were sourced from commercial vendors. Further information on the cell lines used in this study, including their source and molecular profiling datasets can be found on cellmodelpassports.sanger.ac.uk and in Supplementary Table 2. Cell lines used in this study were: AU565, BT-20, BT-474, BT-483, BT-549, CAL-120, CAL-148, CAL-51, CAL-85-1, CAMA-1, COLO-824, DU-4475, EFM-19, EFM-192A, EVSA-T, HCC1143, HCC1187, HCC1395, HCC1419, HCC1428, HCC1500,HCC1569, HCC1599, HCC1806, HCC1937, HCC1954, HCC202, HCC2157, HCC2218, HCC38, HCC70, HDQ-P1, Hs-578-T, JIMT-1, MCF7, MDA-MB-157, MDA-MB-175-VII, MDA-MB-231, MDA-MB-330, MDA-MB-361, MDA-MB-415, MDA-MB-436, MDA-MB-453, MDA-MB-468, MFM-223, MRK-nu-1, OCUB-M, T47D, UACC-812, UACC-893, ZR-75-30, C2BBe1, CaR-1, CCK-81, CL-11, COLO-205, COLO-320-HSR, COLO-678, CW-2, DiFi, GP5d, HCC2998, HCT-116, HCT-15, HT-115, HT-29, HT55, KM12, LoVo, LS-1034, LS-123, LS-180, LS-411N, LS-513, MDST8, NCI-H508, NCI-H716, NCI-H747, RCM-1, RKO, SK-CO-1, SNU-1040, SNU-175, SNU-407, SNU-81, SNU-C1, SNU-C2B, SNU-C5, SW1116, SW1417, SW1463, SW48, SW620, SW837, SW948, T84, AsPC-1, BxPC-3, CAPAN-1, CAPAN-2, CFPAC-1, DAN-G, HPAC, Hs-766T, HuP-T3, HuP-T4, KP-1N, KP-2, KP-3, KP-4, MIA-PaCa-2, MZ1-PC, PA-TU-8902, PA-TU-8988T, PANC-02-03, PANC-03-27, PANC-04-03, PANC-08-13, PANC-10-05, PL4, PSN1, SU8686, SUIT-2, SW1990, YAPC. |
| Authentication | To prevent cross-contamination or misidentification, all cell lines were profiled using a panel of 94 SNPs (Fluidigm, 96.96 Dynamic Array IFC). Short tandem repeat (STR) analysis was also performed, and cell line profiles were matched to those generated by the cell line repository. |
| Mycoplasma contamination | All cell lines are routinely tested for mycoplasma and are negative for mycoplasma |
| Commonly misidentified lines (See ICLAC register) | All cell lines have been manually curated to remove any commonly misidentified cell lines, and SNP and STR authenticated to ensure their identity. |

# Animals and other organisms

Policy information about studies involving animals; ARRIVE guidelines recommended for reporting animal research

| | |
|---|---|
| Laboratory animals | NOD/SCID mice |
| Wild animals | *Provide details on animals observed in or captured in the field; report species, sex and age where possible. Describe how animals were caught and transported and what happened to captive animals after the study (if killed, explain why and describe method; if released, say where and when) OR state that the study did not involve wild animals.* |
| Field-collected samples | *For laboratory work with field-collected samples, describe all relevant parameters such as housing, maintenance, temperature, photoperiod and end-of-experiment protocol OR state that the study did not involve samples collected from the field.* |
| Ethics oversight | Animal procedures were approved by the Italian Ministry of Health (authorization 806/2016-PR). |

Note that full information on the approval of the study protocol must also be provided in the manuscript.

