## [Peer Review File · Nature]

Manuscript Title: Effective drug combinations in breast, colon and pancreatic cancer cells

Reviewer Comments & Author Rebuttals

Reviewer Reports on the Initial Version:

Referee #1

Jaaks, Coker et al. present a combinatorial dataset of drug sensitivity of 2,025 two-drug combinations in 128 cell lines. The scale of this dataset is impressive and represents the largest combinatorial dataset to date. The data together with the accompanying website tool for exploration make this a very useful resource for the scientific community.

The main novel findings of this study are that synergy in combinations is context-dependent and that biomarkers of synergies can be identified and are different from biomarkers for single-agent responses. Finally, the data highlights that targeted agents are more synergistic than chemotherapeutics.

The authors follow-up one of the identified synergies, TOP1 and CHECK1, with validation performed both in vitro and in vivo.

The manuscript is well written and references are appropriately cited.

Major points:

One of the main conclusions of the paper is that the synergy in combinations is context-dependent. However, it seems that the variation across contexts could be very sensitive to how the authors threshold which combination/cell line pairs show synergy. In fact, in the validation screen, the "precision" of the synergy calling was low with less than half of called synergies being validated. In addition, the single-agent responses could explain some of the context variation of the synergies. It could be helpful if the authors show a more clear quantification of the synergy and an analysis of how their synergy calls relate to single-agent responses.

For the biomarker analysis, the authors conclude that the biomarker associations for combinations are more 'distanced' to the target in the biological network compared to single-agent responses. Could this be explained by a higher rate of false-positive associations because of the relatively lower signal-to-noise ratio for measuring synergy vs single-agent responses? It would be helpful if the authors could show an analysis to prove that this is not the case.

The authors identify that the synergy of TOP1 and CHEK1/2 is particularly strong MSS and KRAS mutant colorectal cancer cell lines. They provide very nice in vitro and in vivo confirmation of the synergy. However, the novelty of the finding is in identifying specific contexts where the synergy occurs. It would be valuable to shed some light on the underlying molecular mechanism as to why this synergy occurs in certain contexts, for example, by performing gene-expression or proteomics profiling in sensitive and insensitive cell lines.

Referee #2

In this manuscript, Jaaks, Coker, et al. report a large-scale cancer cell line screening effort to discover effective drug combinations with selective anti-cancer efficacy. The screen included 128 established cell lines from 3 cancer types (breast, colorectal, and pancreatic cancer). Compounds were selected based on approved therapies for each cancer type and the investigators' choice of investigational or targeted therapies. To minimize the number of required assay wells, an "anchor"

screening strategy was employed. Two doses of a reference compound were combined with 7 doses of each test compound. After optimizing seeding density, the screen was performed using 1536-well plates with a CellTiter-Glo readout. Drug combinations were scored based on shift in IC50 or Emax beyond the expected additive effect. Active combinations were re-tested in a smaller cell line set and several two-drug combinations were selected for biological validation studies.

The manuscript is well-written and emphasizes several interesting high-level conclusions including that drug synergy is relatively rare and occurs most frequently for compounds paired with CHEK1, BCL2, or MTOR pathway inhibitors. Strengths of the study include the use of stringent QC metrics, integration of robust cell line molecular feature sets, and the creation of a web portal to view the results. However, the specific vignettes highlighted in the paper may have limited impact. The two categories of promising drug combinations pursued with validation studies (CHEK1 and BCL2-related combinations) have been previously proposed. Multiple CHEK1 inhibitor combinations have been or are currently in clinical development in combination with chemotherapy agents. For example, a phase 1 trial combining AZD-7762 and irinotecan was terminated (NCT00473616). BCL2 is a common hit in combinatorial genetic screens and navitoclax-based combinations have also been proposed.

The greatest impact of the work could be use of the resource to discover novel combinations or in influencing the design of future screening efforts. However, the landscape figure heatmaps are very dense and convey limited information in print form. Interactive versions of these plots could be more informative. The GDSC2 website screenshots look promising but the website was not made available for evaluation by reviewers (as far as I can tell).

The discussion section would benefit from more reflection and critical appraisal of the initial dataset. What should be done next? Screen more models or more drug combinations or change the overall approach entirely? Did the 2x7 dose matrix strategy prove optimal? Could a future screen testing a smaller number of anchor drugs versus entire compound libraries be useful given the rarity of synergy partners?

Specific comments:

1. Selection of anchor dose: The anchor dose screening strategy is appealing to limit the number of required assay wells. In the screen, the low/high-dose anchor concentrations were set at the cell lineage level and activity against individual cell lines varies per Ext. Fig. 2d. Interestingly, the synergy concordance between the low/high anchor doses was limited, suggesting the choice of anchor dose is quite important. Would it be advantageous to select an optimized anchor dose for each cell line individually?

2. Screen QC: Appropriate metrics were used to evaluate plate performance, with about 30% of plates failing CV or Z-factor thresholds. Were replicates used in the primary combination screen? The replicate strategy should be described more explicitly. Were any plate failures repeated (30% seems like a large amount of data to filter out)? Did failures occur at random or were they enriched for specific cell lines or compound source plates? Can any failures be explained by suboptimal positive control activity against specific cell lines reducing the Z-factor?

3. Synergy and cell growth metrics: Extended data Fig. 1f is quite helpful in depicting the analysis methodology. Were any synergy pairs confirmed using a full dose response matrix and multiple synergy score metrics (e.g., Loewe or ZIP models)? This seems important to understand the full range of concentration-dependent effects. For example, for a given compound pair with evident IC50 shift, does synergy result from lowering the dose of both agents (dose sparing) or greater efficacy? It may be difficult to draw clear conclusions with the initial 2x7 data alone. Also, were any surprising patterns identified (e.g., synergy and antagonism for the same compound pair in different models or across different dose ranges)? Finally, were day 0 viability measurements taken to infer growth inhibition?

4. Recovery of known drug combinations: Several examples of known targeted therapy + chemotherapy combinations were identified. Were any existing effective chemotherapy combinations recovered (e.g., 5-FU and oxaliplatin for colorectal cancer)?
5. Potential for false positives in vitro: Were any known false positive combinations that have failed to replicate in vivo (e.g., IGF1R inhibitors + MEK inhibitors; PMID26479923) recovered by the screen? Given this risk with in vitro screens in general, does every combination need to be evaluated in vivo prior to clinical translation?
6. Biomarker associations: Why binarize the continuous gene expression values? For high-priority hits, would a full synergy matrix of two compounds provide a more robust profile for biomarker analysis?
7. Plots: Multiple plots are information-rich but convey limited information in print form given the small size (e.g., the drug labels in Fig. 1b and unlabeled strong outliers in Fig. 3a/Fig. 4a/Ext. Fig. 4c-d). This could be improved with labeling of additional points, addition of interactive plots online, and potentially showing important plot regions at higher magnification. It would also be helpful for the processed plot source data to be available.
8. Genetic target validation: A single siRNA pool appears to have been used to knockdown CHEK1 which raises the risk for an off-target effect. The authors did employ reagents purportedly designed to minimize off-target seed effects. How effective is the knockdown efficiency against CHEK1? Also, how much does the irinotecan IC50 shift in response to CHEK1 knockdown? This result would also be stronger if reproduced using multiple independent reagents/orthogonal methods or if rescued using a knockdown-resistant cDNA.
9. In vivo efficacy experiments: The difference in tumor growth curves appears to be primarily driven by irinotecan treatment. The SNU-81 regrowth study is quite promising but would benefit from additional mechanistic studies. Are there any PD markers that correlate with this response? Are cell death pathways uniquely activated in vivo by the combination? Does any viable tumor tissue remain?
10. Compound verification: The authors state that the compound identity was confirmed by LC-MS. Was there a compound purity threshold as well? Can the compound source and purity information be reported with the resource?
11. GDSC2 website: The website should be made available to reviewers if considered part of the resource.

Referee #3

A well written largely descriptive manuscript documenting a monumental amount of repetitive experiments (>300,000 combination experiments) done to a high quality which would serve as a data base for researchers to use for a long time. This builds on other publications on combination and cancer done by group *Nat Commun.* 2019 Jun 17;10(1):2674. Conducting experiments that create a large volume of data is beneficial to the research community and gives researchers access to large amounts of data but also sophisticated analysis to go along with it. Good examples include TCGA and depmap initiatives/portals. Often such databases have gene mutation (whole genome/exome), gene expression (RNA seq), CRISPR and to a lesser extent proteomic data. The data generated in this paper will fall shy of such large data bases not because of a lesser effort of conducting such a large volume of experiments but rather a lesser diversity of output i.e. grades of synergy of two way combinations being the major output, which will be dwarfed when compared to whole genome/RNA seq/CRISPR data. The analysis done reflects what data the researchers have had available to them, i.e. 2 drug combinations and is mainly descriptive. Also the description of a CHK1 inhibitor causing synergy with irinotecan or a BCL2 inhibitor in combination with inhibitors of TOP1 does not add significant knowledge to what is already realised in smaller hypothesis testing

experiments. However a web-based portal (GDSC2) making this data accessible will be valuable to cancer researchers the world over.

Specific questions

1. What was the basis for picking the nine molecular baskets representing specific molecular subgroups (Line 274, Figure 1A)?, While markers like MSS and MSI are important biomarkers in the diseases concerned, their predictive use thus far has been in immunotherapy which was not evaluated in this manuscript.
2. Small molecule drugs are known to have significant effects on multiple targets e.g. dasatinib inhibits ABL, SRC, C-KIT, SRC etc. How was this accounted for when calculating the shortest finite network distance (line 313, Figure 3b)?
3. In the discussion the authors have mentioned limitations of the data which include lack of stromal effects, impracticality of studying organoids in such high throughput experiments, which are true. Worth adding that effects on a panel of 'normal' non-cancer cells in selected combinations would also be of benefit as the translation of combination therapies defined pre-clinically is almost universally challenged by the inability of being able to combine the drugs without causing excessive clinical toxicity. The examples of CHK1 inhibitor + irinotecan cited as an example in the manuscript is a good case, as more than a decade of clinical research into combinations of CHK inhibitors and chemotherapy has not yielded a registration primarily due to excess normal tissue toxicity seen in clinical trials.

Author Rebuttals to Initial Comments:

Response to referees' comments

We thank the referees for their positive and constructive feedback on our manuscript. We believe that we have addressed their concerns and the resulting manuscript is improved. Here we provide a brief summary of the major revisions followed by a point-by-point response to individual referees' comments, together with accompanying additional experimental data. Our responses are in bold text.

Summary of major revisions to the manuscript:

- 1. We have re-performed our STR authentication of all cell line expansions used to generate the data and have identified three problematic cell lines (one from pancreas, two from colon). All associated data have been removed from the manuscript and single agent and combination responses have been refitted. We have updated the manuscript throughout and note that the validation rate of synergy calls has improved.**

2. Our drug combination data is now available on our website GDSC Combinations (<https://gdsc-combinations.depmap.sanger.ac.uk/>). The website is currently password protected (username: trinity; password: dodgethis) but will be made publicly available and password protection removed at the time of publication of our manuscript. We invite the referees to explore the data.

3. To strengthen the follow-up studies on CHEK1i+TOP1i in colon we have conducted additional in vitro experiments, analysed collected xenograft tumour samples and run additional biomarker analyses to refine the populations with most combination benefit. We have found that: 1) knock-down of CHEK1 but not CHEK2 increases sensitivity (reduced IC50) of colon cells to SN-38 (TOP1i); 2) CHEK1i+TOP1i treatment in vivo leads to reduced proliferation, increased cell death and more pronounced induction of DNA double strand breaks compared with single-agent TOP1i; and 3) two previously unreported colon subpopulations, MSS colon and KRAS-TP53 double mutant colon, show particular sensitivity to the combination. Hence, while CHEK1 inhibitors in combination with DNA damaging agents have previously been reported with modest clinical activity, we believe that we present a strong case for targeting two specific patient populations warranting further preclinical follow-up in the future.

Referee #1 (Remarks to the Author):

Jaaks, Coker et al. present a combinatorial dataset of drug sensitivity of 2,025 two-drug combinations in 128 cell lines. The scale of this dataset is impressive and represents the largest combinatorial dataset to date. The data together with the accompanying website tool for exploration make this a very useful resource for the scientific community.

The main novel findings of this study are that synergy in combinations is context-dependent and that biomarkers of synergies can be identified and are different from biomarkers for single-agent responses. Finally, the data highlights that targeted agents are more synergistic than chemotherapeutics.

The authors follow-up one of the identified synergies, TOP1 and CHECK1, with validation performed both in vitro and in vivo.

The manuscript is well written and references are appropriately cited.

We thank the referee for recognising the resource element of the manuscript, as well as the value of synergy and biomarker analyses.

Major points:

One of the main conclusions of the paper is that the synergy in combinations is context-dependent. However, it seems that the variation across contexts could be very sensitive to how the authors threshold which combination/cell line pairs show synergy. In fact, in the validation screen, the “precision” of the synergy calling was low with less than half of called synergies being validated. In addition, the single-agent responses could explain some of the context variation of the synergies. It could be helpful if the authors show a more clear quantification of the synergy and an analysis of how their synergy calls relate to single-agent responses.

We agree with the referee that the synergy classification and context dependency of synergistic effects are important topics of our manuscript. We acknowledge that multiple factors might influence which specific combination-cell line pairs will be classified as synergistic, including the synergy thresholds and the chosen concentrations of the single-agents. Nonetheless, as discussed here, through further analysis we confirm that the context dependence is independent of the synergy threshold applied.

To identify synergistic effects we opted for classifying combination-cell line pairs based on shifts in efficacy and potency, an approach also recently described by Meyer et al. ¹. This is an intuitive way of classifying based on shifts in viability (E_{max}) or drug concentration required to achieve a half-maximal viability reduction (IC_{50}) (Extended Data Figure 1g), both of which could have clinical relevance to enhance tumour cell killing (reducing residual disease tissue), and for increased sensitivity to drugs and possible dose reductions, respectively. We applied stringent synergy thresholds of $\Delta E_{max} \geq 20\%$ viability or $\Delta IC_{50} \geq 3$ (=8-fold concentration; see methods section for more details) to ensure that only the strongest effects were classified as synergistic. To directly address how varying the synergy thresholds impacts the landscape of synergy, we have now applied three sets of synergy thresholds and repeated our analyses. Along with the original synergy filters, we have applied one pair of filters that is more stringent ($\Delta E_{max} \geq 30\%$ viability or $\Delta IC_{50} \geq 4$ (=16-fold)) and one pair of filters that is less stringent ($\Delta E_{max} \geq 10\%$ viability or $\Delta IC_{50} \geq 2$ (=4-fold)).

Firstly, as expected, the frequency of synergistic combination-cell line pairs increases as we reduce the stringency of synergy thresholds, reaching over 10% when applying the least stringent thresholds (see table below). We note, however, that the relative frequency of synergy between cancer types is maintained independent of threshold, suggesting that the threshold used does not bias synergy calls within a specific tissue.

Tissue	$\Delta E_{max} \geq 10\%$ or $\Delta IC_{50} \geq 2$ (less stringent)	$\Delta E_{max} \geq 20\%$ or $\Delta IC_{50} \geq 3$ (original thresholds)	$\Delta E_{max} \geq 30\%$ or $\Delta IC_{50} \geq 4$ (more stringent)
All tissues	12.3%	5.2%	2.3%
Breast	11.2%	4.4%	1.9%
Colon	13.3%	5.4%	2.3%
Pancreas	14.5%	7.2%	3.5%

Furthermore, we find that within the 121 pan-tissue combinations the relative tissue-specific synergy rates are well-preserved using the different synergy thresholds. This is illustrated by looking at the 27 pan-tissue combinations that showed $\geq 20\%$ synergy in at least one tissue when applying our current synergy thresholds of $\Delta E_{max} \geq 20\%$ or $\Delta IC_{50} \geq 3$ (see plot below). Additionally, the ranking of combinations by synergy is well preserved within each tissue. For example, crizotinib+navitoclax remains amongst the combinations with the highest frequency of synergy in pancreas, irrespective of the threshold applied. These results have been included in the manuscript as Extended Data Figure 3b.

We find that relationships between synergy and biomarker associations are also well-preserved when applying different synergy thresholds. For instance, we reported that synergy for sapitinib (EGFR, ERBB2/3) and JQ1 (BRD2/3/4/T) exclusively occurred in cell lines with ERBB2 copy number gain. With the exception of the most permissive synergy threshold, which results in synergy in two ERBB2 wt cell lines, this holds true when applying different synergy thresholds (see below). Another biomarker example is BRAFi with EGFRi which exclusively showed synergy in BRAF mut colon cancer cell lines and is considered a gold standard biomarker of combination response. This biomarker association is identified in all cases when applying different synergy thresholds, however the most permissive threshold not only classifies two additional BRAF mut cell lines as synergistic for dabrafenib+afatinib, it also classifies one BRAF wt cell line as synergistic. These examples demonstrate that our synergy thresholds of $\Delta E_{max} \geq 20\%$ and $\Delta IC_{50} \geq 3$ strikes a balance between finding expected associations without misclassifying cell lines (data not shown).

When examining recall, precision and accuracy (F-score) from our validation screen, we observed that recall decreases with increased synergy threshold stringency, whereas precision and F-score are fairly consistent between synergy filters (see plot below). Please note that the precision and in turn the F-score, which takes recall and precision into account, have improved upon refitting the whole dataset (precision now: 0.77 for breast, 0.62 for colon, 0.56 for pancreas; precision previously: 0.37-0.4 ; F-scores now: 0.7 for breast, 0.62 for colon, 0.65 for pancreas; F-scores previously: 0.49-0.55). Together, these analyses demonstrate that our chosen synergy thresholds of $\Delta E_{max} \geq 20\%$ and $\Delta IC_{50} \geq 3$ results in a reasonable synergy rate, that context-specificity of synergy is largely conserved independent of the synergy threshold applied (albeit at different absolute synergy rates), and that this threshold results in a consistent synergy classification across the original and validation screens.

To address how single-agent activity relates to synergy, we compared anchor and library responses in synergistic and non-synergistic measurements. We observe that synergy is associated with a greater anchor effect on cell viability (see plots below; Student's t-test. ****= $p < 0.0001$). This is observed at both low and high anchor concentrations, suggesting that synergy requires at least some target engagement from the anchor compound. Furthermore, the viability effect of anchor compounds is less than library compounds. We thank the referee for suggesting useful analysis which has been added to the manuscript as Extended Data Figure 3a.

Synergistic populations:

Low anchor: 52 - 86% interquartile (IQR)

High anchor: 69 - 92% IQR

Library: 53 - 80% IQR

Additionally, we have looked into whether single-agent effects and combination responses are strongly correlated. As seen below, both anchor viability (both anchor concentrations plotted) and library E_{max} are not clearly correlated with ΔE_{max} and ΔIC_{50} ($R = -0.051$ to 0.16 , colour indicates density from low density in blue to high density

in yellow), demonstrating that the synergistic effects cannot simply be attributed to efficacy of the single-agents.

Collectively, these results suggest that synergy is associated with single-agent drug response, but that synergistic effect size is not proportional to the single-agent effect size.

For the biomarker analysis, the authors conclude that the biomarker associations for combinations are more 'distanced' to the target in the biological network compared to single-agent responses. Could this be explained by a higher rate of false-positive associations because of the relatively lower signal-to-noise ratio for measuring synergy vs single-agent responses? It would be helpful if the authors could show an analysis to prove that this is not the case.

Referee 1 raises a good question regarding false positives for biomarkers. We have applied an FDR filter (based on q value correction²) of $\leq 5\%$, which will control for false discovery of associations and thus minimise the number of false positives. However, as the first such large-scale drug combination biomarker analysis we have no way of knowing the ground truth in terms of genuine combination biomarkers, and as such we cannot calculate the number of false positives.

However, by randomly assigning biomarkers to combinations to simulate entirely false positive associations, we can assess if these network effects shown in the original Figure 3b (shown below, left - updated version in new Extended Data Figure 5b) are still seen. These randomised data are now displayed in Extended Data Figure 5d (shown below, right). These plots show that for combinations the target itself is less likely to be the biomarker than for single agents (shortest distance = 0), but for our simulated false positive associations, there is no such difference for shortest distance = 0 between target-biomarker distances seen for single-agent and combination biomarkers.

Reactome results (non-randomised data)

Reactome results (randomised data)

This suggests that the results reported in the original Figure 3b are not due to false positive associations and that the observation is due to true positive, or genuine, biomarker-feature associations.

To further support the robustness of the ‘distanced’ observation, we have now also used an alternative, larger interactome to calculate network distances, IntAct³: this IntAct interactome contains approximately 9,000 more protein nodes and more than 85,000 more interactions than the previous Reactome interactome. The results of this analysis are now shown in the new Extended Data Figure 5a (shown below, left), which illustrate that the observation is not interactome-specific. The previous version of this analysis, using the Reactome interactome, has now been moved to Extended Data Figure 5b. New Extended Data Figure 5c (shown below, right) shows the results of randomly assigned biomarkers based on the IntAct interactome, supporting the finding that the

‘distanced observation’ is a feature of the genuine biomarker associations. The key area for comparison between the plots is highlighted here with a red box.

IntAct results (non-randomised data)

IntAct results (randomised data)

The methods section, text of the manuscript, Figure 3 and Extended Data Figure 5 have now been updated to reflect these updated analyses.

The authors identify that the synergy of TOP1 and CHEK1/2 is particularly strong in MSS and KRAS mutant colorectal cancer cell lines. They provide very nice in vitro and in vivo confirmation of the synergy. However, the novelty of the finding is in identifying specific contexts where the synergy occurs. It would be valuable to shed some light on the underlying molecular mechanism as to why this synergy occurs in certain contexts, for example, by performing gene-expression or proteomics profiling in sensitive and insensitive cell lines.

We agree with the referee that it would be valuable to understand what distinguishes synergistic and non-synergistic cell lines. We identified MSS status as a significant biomarker of response, but further stratification of sensitive models or patients could be beneficial. Subsequently, we have performed additional focused analyses that have shed further light on the context and mechanism of this synergy.

Following on from our initial observation, we confirm that MSS colon cell lines show a significantly higher combination ΔE_{max} (left - $p = 0.00141$ with Bonferroni correction) and ΔIC_{50} (right - $p = 0.000786$ with Bonferroni correction) than the MSI population. TOP1+CHEK1 combinations have not previously been associated with MSS colon cancer, and thus we believe that the identification of this specific context of combination response is novel.

Response stratification of AZD7762 with camptothecin by microsatellite stability

In studies using small numbers of cell lines, KRAS mutations have been linked to activity of CHEK1 inhibitors⁴ and TP53 mutations have been reported as a biomarker of response to TOP1-CHEKi combinations⁵. We find that within the KRAS mutant colon molecular basket, TP53 mutated cell lines show a significantly higher combination ΔE_{max} (left - $p < 0.001$ with Bonferroni correction) and ΔIC_{50} (right - $p < 0.001$ with Bonferroni correction) than the TP53 wild type population. The sensitivity of this KRAS-TP53 double-mutant population has not been described in the literature, and thus represents a second novel context for combination response.

Response stratification of AZD7762 with camptothecin by TP53 mutation in KRAS mutant cell lines

ΔE_{max}

ΔIC_{50}

For completeness, we have also compared combination responses between KRAS wt and KRAS mutant cells within the MSS colon population: we see a non-significant difference in response in terms of both ΔE_{max} (below left, $p = 0.758$ with Bonferroni correction) and ΔIC_{50} (below right, $p = 0.861$ with Bonferroni correction).

Response stratification of AZD7762 with camptothecin by KRAS mutation in MSS cell lines

ΔE_{max}

ΔIC_{50}

In addition to the results shown, we retrospectively performed extensive further analysis of these two populations (MSS colon and KRAS-TP53 double mutant colon) to attempt to identify the mechanisms behind the responses, for example additional statistical tests with active/inactive SPEED pathways⁶ and known gene dependencies of the cell lines^{6,7}: however, these analyses did not provide additional insight into the mechanisms behind sensitivity. We have also conducted a thorough literature review to identify potential mechanisms. A 2014 clinical trial⁶⁻⁸ observed complete and durable response to a CHEKi-topoisomerase inhibitor combination in a patient that was

attributed to a mutation in their RAD50 gene: although 17/46 of our colon cell lines had a mutation in RAD50⁹, we did not see an association between mutation of this gene and response to the combination in our screen.

Furthermore, to shed light on the underlying mechanism of synergy we have analysed LS-1034 xenograft tumours collected 72h after start of in vivo drug treatment. Samples from untreated as well as single agent and combination treated tumours were FFPE embedded, cut and IHC immunoreactivity for Ki67 (proliferation index), caspase-3 (apoptosis index) and phospho-H2AX (DNA double strand breaks) were analysed. We analysed data for 1-3 tumours per treatment arm and 2-10 (Ki67 and phospho-H2AX) or 3-5 optical fields (caspase-3). As seen in the plot below, 72h treatment of LS-1034 xenografts with irinotecan in combination with rabusertib leads to a significantly lower proliferation index (Ki67; left), a higher rate of cells staining positive for DNA double strand breaks (phospho-H2AX; middle) and conversely a higher rate of apoptotic cells (caspase-3; right) compared to irinotecan monotherapy. The accumulation of more genotoxic DNA damage likely explains the more pronounced tumour growth inhibition observed in combination-treated tumours. Interestingly, ectopic introduction of mutant KRAS in TP53 knock-out cells has been shown to induce DNA replication stress, which results in DNA double-strand break accumulation¹⁰. With this in mind, it is tempting to speculate that the endogenous co-occurrence of TP53 and KRAS mutations engenders a context of DNA destabilization that makes tumours particularly susceptible to DNA-damaging agents, such as the combination of CHEK1 and topoisomerase inhibitors. These results have been included in the manuscript.

We believe that our finding that the specific combination of TOP1+CHEK1 inhibition is particularly active in MSS and/or KRAS-TP53 double mutant colon cancer is noteworthy due to its clear route to the clinic. CHEK inhibitors are well-tolerated in patients as single agents and remain in clinical development¹¹⁻¹⁵. Whilst combinations of

CHEKi and DNA damaging agents have shown to be effective in treating very specific cancer types such as non-small cell lung cancers or leukaemias^{16–18}, low activity and toxicity issues in other clinical trials, notably those that do not stratify patients by cancer type and instead study ‘advanced solid cancers’, have been reported^{11,19–26}. The use of CHEK1-selective inhibitors may also provide a larger therapeutic window by reducing on- and off-target toxicity. As we describe, synergy and beneficial responses to drug combinations are extremely context-specific, and therefore we propose that patient stratification is likely to be key to clinical success of any given combination. Hence, our identification of the combination response of CHEK1i+TOP1i is particularly exciting as we identify two patient subpopulations, namely MSS colon cancers and KRAS-TP53 double mutant colon cancers, in which this response is most likely to be beneficial.

Referee #2 (Remarks to the Author):

In this manuscript, Jaaks, Coker, et al. report a large-scale cancer cell line screening effort to discover effective drug combinations with selective anti-cancer efficacy. The screen included 128 established cell lines from 3 cancer types (breast, colorectal, and pancreatic cancer). Compounds were selected based on approved therapies for each cancer type and the investigators’ choice of investigational or targeted therapies. To minimize the number of required assay wells, an “anchor” screening strategy was employed. Two doses of a reference compound were combined with 7 doses of each test compound. After optimizing seeding density, the screen was performed using 1536-well plates with a CellTiter-Glo readout. Drug combinations were scored based on shift in IC50 or Emax beyond the expected additive effect. Active combinations were re-tested in a smaller cell line set and several two-drug combinations were selected for biological validation studies.

The manuscript is well-written and emphasizes several interesting high-level conclusions including that drug synergy is relatively rare and occurs most frequently for compounds paired with CHEK1, BCL2, or MTOR pathway inhibitors. Strengths of the study include the use of stringent QC metrics, integration of robust cell line molecular feature sets, and the creation of a web portal to view the results. However, the specific vignettes highlighted in the paper may have limited impact. The two categories of promising drug combinations pursued with validation studies (CHEK1 and BCL2-related combinations) have been previously proposed. Multiple CHEK1 inhibitor combinations have been or are currently in clinical development in combination with chemotherapy agents. For example, a phase 1 trial combining AZD-7762 and irinotecan was terminated (NCT00473616). BCL2 is a common hit in combinatorial genetic screens and navitoclax-based combinations have also been proposed.

The greatest impact of the work could be use of the resource to discover novel combinations or in influencing the design of future screening efforts. However, the landscape figure heatmaps are very dense and convey limited information in print form. Interactive versions of these plots could be more informative. The GDSC2 website screenshots look promising but the website was not made available for evaluation by reviewers (as far as I can tell).

The discussion section would benefit from more reflection and critical appraisal of the initial dataset. What should be done next? Screen more models or more drug combinations or change the overall approach entirely? Did the 2x7 dose matrix strategy prove optimal? Could a future screen testing a smaller number of anchor drugs versus entire compound libraries be useful given the rarity of synergy partners?

We thank the referee for commenting on the resource value and quality of the drug combination datasets. The new drug combination website GDSC Combinations (<https://gdsc-combinations.depmap.sanger.ac.uk/> (username: trinity; password: dodgethis)) has been made available to the referees with this resubmission and now contains interactive versions of the data.

We acknowledge that CHEK1 inhibitor combinations have been described in the past, often with variable efficacy in clinical trials. However, many of these trials were performed with less selective drugs (for example AZD7762) and did not include biomarkers of combination response to stratify patient populations. Hence, we would like to highlight that we identified two specific populations, MSS colon and KRAS-TP53 double mutant colon (see below), that showed higher sensitivity to CHEK1i+TOP1i in our study and follow-up experiments. We believe that improved patient stratification and improved compound selectivity are key to demonstrating clinical benefit for CHEK1 inhibitor combinations, as demonstrated by others¹⁶⁻¹⁸.

Response stratification of AZD7762 (CHEK1/2) with camptothecin (TOP1)

Based on microsatellite status

Based on TP53 mutation in KRAS mut cell lines

We thank the referee for their suggestion of including additional appraisal and reflection on our findings. We refer the referee to the discussion section of the manuscript and summarise some of the key points included below:

- a. We discuss the value of evaluating both sensitivity (deltaIC50) and efficacy (deltaEmax), and how this could lead to the identification of combinations leading to dose reduction, improved efficacy, or both, relative to single agents.
- b. We propose that similar screens in additional cancer types would be worthwhile, especially given the context specificity of combinations tested.
- c. Focused studies using more complex culture models, or screening of higher-order combinations, which reflect the use of higher-order chemotherapy combinations in use in the clinic, could enhance and extend our findings.
- d. We suggest that the testing of selected combinations in normal, non-cancer cell lines may help to estimate clinical toxicity.
- e. We mention that in addition to our analysis of drug synergy, these data may be used to identify combinations of drugs, each of which alone is independently active in a certain subset of patients, and when combined could be more effective at the patient population level.
- f. We also point out that our finding that drugs with weak or modest single-agent activity, and those separated by 1 or 2 nodes in a protein-protein interaction network, are most likely to yield a synergistic interaction could be used to improve design of future screens by nominating drugs most likely to be synergistic and reducing the combinatorial search space investigated.
- g. Similarly, we suggest that our data could improve computational approaches for predicting effective drug combinations in different context, which are currently underpowered due to a lack of comprehensive training datasets, with potential to identify effective combinations far beyond the drugs and cell lines tested here.
- h. We additionally mention the limitations of our study, including the choice of drugs and the screening concentrations used, as well as the limitation of in vitro cancer cell lines.

We believe that these points and others improve the manuscript and seek to address the points raised by the reviewer.

Specific comments:

1. Selection of anchor dose: The anchor dose screening strategy is appealing to limit the number of required assay wells. In the screen, the low/high-dose anchor concentrations were set at the cell lineage level and activity against individual cell lines varies per Ext. Fig. 2d. Interestingly, the synergy concordance between the low/high anchor doses was limited, suggesting the choice of anchor dose is quite important. Would it be advantageous to select an optimized anchor dose for each cell line individually?

For experimental and logistical reasons, we chose to screen the same two anchor concentrations across all cell lines of a specific cancer type. These were selected from existing monotherapy data and literature, information on drug C_{max}, and further optimised in a pilot screen on 9-13 cell lines per tissue to modulate their target while providing a range of responses across the cell lines (see methods section on compounds). Nevertheless, the referee's question is very interesting, and we sought to address whether we could have chosen one anchor concentration per cell line while retaining synergies observed across all anchor-library pairs screened for a cell line. We calculated for all 3,199 anchor-cell line pairs in the screen whether synergy for their libraries was found at high, low or both anchor concentrations. Anchor-cell line pairs in breast have been screened with up to 51 library compounds, anchor-cell line pairs in colon and pancreas with up to 26 library compounds. Having a closer look at all 361 (11.3%) of anchor-cell line pairs that showed synergy with at least five library compounds, we observed that which anchor concentration leads to synergy is library dependent in 53.5%. For example, the cell line CAL-120 yielded synergy for 24 out of 51 library compounds paired with the anchor navitoclax. While synergy was found at both anchor concentrations for 11 library compounds, it exclusively occurred at high or low anchor concentrations for 5 or 8 library compounds, respectively. As these results show, limiting the anchor concentration to one optimised dose per cell line could result in loss of synergistic signals. We would also like to point out that optimisation of anchor doses for each cell line individually might not always be possible, for example if a cell line is insensitive to a compound and/or there is no pharmacodynamic marker of target engagement in cells, and would be logistically extremely challenging across such a large panel of cell lines.

2. Screen QC: Appropriate metrics were used to evaluate plate performance, with about 30% of plates failing CV or Z-factor thresholds. Were replicates used in the primary combination screen? The replicate strategy should be described more explicitly. Were any plate failures repeated (30% seems like a large amount of data to filter out)? Did failures occur at random or were they enriched for specific cell lines or compound source plates? Can any failures be

explained by suboptimal positive control activity against specific cell lines reducing the Z-factor?

We thank the referee for asking these important questions around quality control and for pointing out that the replicate strategy was not clearly described. The methods section “Assay plate quality control” has been revised to address this.

A single replicate of the 7-point combination dose response was performed in the primary screen. However, because of their importance when calculating synergy effects, we performed 5 technical replicates of the single-agent anchor (at each concentration) and 4 technical replicates of the single-agent library dose responses on each plate. Each cancer-type specific screen included 4-5 cell lines for which at least three independent biological replicates were collected for all combinations over the duration of screening. Each biological replicate was composed of three technical replicate plates. These data were used to assess reproducibility of the data within a dataset, and over time, and the correlation of single-agent and combination response metrics for biological replicates was high ($r > 0.6$, $p\text{-value} < 0.05$) (Extended Data Fig. 2c).

We designed the screening plate such that we had sensitivity to detect outlier plates that had unusual variability, which contributes to the 30% plate failure rate. For instance, multiple positive ($n = 68$) and negative ($n = 132$) control wells were distributed throughout the 1536-well screening plate, including in the corner of plates, to be sensitive to edge effects (note that the outer two edge wells are not used). We also note that in rare instances technical, operational or environmental issues led to the failure of whole plate runs, contributing to the plate failure rate. This would have included some plates which would have passed based on CV and Z-factor plate metrics. Notably, if a plate had to be failed due to failing CV or Z-factor thresholds, we repeated the plate where possible, resulting in $\geq 96\%$ of dataset completeness for all three tissues (breast: 96.5%, colon: 99.8%, pancreas: 99%). We note that failures were not enriched for specific compound plate sets, which each typically contained 26 anchors screened against two libraries.

For operational reasons the screening was performed in batches with all plates for a single model typically being screened within a period of a few months. Clustering of the failures by specific compound source plates and/or cell lines is confounded by this batching and associations with plate dates corresponding to technical issues affecting either dispensing, dosing or reading. We did not find any major parameters like seeding density or growth property (adhesion or suspension) to be enriched for cell line failure.

Regarding suboptimal positive control activity, we used two positive control drugs per plate to mitigate this effect, namely staurosporine and MG132. Nonetheless, we observe that models with weak to moderate sensitivity to either positive control are more challenging to screen than those with strong sensitivity to the positive controls. In some rare instances cells were resistant to both positive controls in which case ‘blank’ wells containing no cells were used when calculating Z-scores.

3. Synergy and cell growth metrics: Extended data Fig. 1f is quite helpful in depicting the analysis methodology. Were any synergy pairs confirmed using a full dose response matrix and multiple synergy score metrics (e.g., Loewe or ZIP models)? This seems important to understand the full range of concentration-dependent effects. For example, for a given compound pair with evident IC₅₀ shift, does synergy result from lowering the dose of both agents (dose sparing) or greater efficacy? It may be difficult to draw clear conclusions with the initial 2x7 data alone. Also, were any surprising patterns identified (e.g., synergy and antagonism for the same compound pair in different models or across different dose ranges)? Finally, were day 0 viability measurements taken to infer growth inhibition?

We thank the referee for appreciating the curve fitting and synergy classification schematics. Our synergy classifications are based on comparisons of the observed combination response with the expected combination response based on Bliss independence. To date, we have not performed analyses based on other synergy score metrics (e.g. Loewe or ZIP models), but appreciate that we or others might explore a wider range of synergy score metrics in the future.

Our screening setup was optimised to detect synergy: anchor and library drug concentrations were chosen to yield weak to moderate responses as monotherapy (see plot below of cell line viability), opening up an observational window to detect synergy (i.e. significant response improvement beyond single agent activity). Hence, our data offer a narrow observational window to detect antagonism, and thus are less sensitive to identifying compound pairs showing antagonism in certain contexts and synergy in others.

The referee asked whether synergies could be confirmed using a full dose response matrix. To investigate this, we independently screened two combinations, tasisib (PI3K inhibitor, β -sparing) combined with trametinib (MEK1/2) or SCH772984 (ERK1/2), in a

7x7 matrix across all three tissues. To compare synergy calls for individual combinations, we pooled synergy calls from both anchor orientations (e.g. taselisib+trametinib and trametinib+taselisib, with the first drug being the anchor), resulting in one synergy annotation per cell line. These were matched with matrix response data. For the matrix data we calculated the average Bliss excess across the full matrix. Bliss excess (i.e. observed combination response - expected combination response with 0.1 corresponding to 10% decrease in viability over the expected) is calculated for each of the 7x7=49 combination response wells, summed up and divided by the number of wells. A positive Bliss excess indicates synergy. Synergy classifications from an anchored format correspond with matrix responses as synergistic cell lines on average show higher Bliss matrix values than non-synergistic cell lines (see plots below; unpaired t-test). Where results vary between screens, this could be the result of non-overlapping screening concentrations and differences between synergy metrics used. For example, the use of average Bliss across the matrix might be relatively insensitive to local synergy effects over a limited range of concentration, or similarly, synergy effects may be underestimated due to the use of non-optimal concentrations within the matrix, whereas categorical synergy classification based on fitted dose response curves can lead to classification errors and loss of information about effects sizes. Hence, while matrix responses can be extended and refined in the future, and further work is required integrate different synergy metrics, we have seen that a minimal matrix design (i.e. anchored format) generally recapitulates combination responses of a full matrix design.

The referee raises an important point on understanding concentration-dependent effects. In our data derived from a 2x7 matrix presented in this manuscript, we use the same anchor and library concentrations for monotherapy and drug combination responses and score synergy based on shifts in IC50 (potency) or Emax (efficacy). We find that while 45.1% of the 9,689 synergistic measurements had large shifts in both, 32.7% and 22.2% of them exclusively showed increased potency or efficacy, respectively, demonstrating their complementary nature. We anticipate that additional data on full (i.e. 7x7) matrix combinations can be used to analyse whether synergistic combinations

can be identified for which doses of both agents could be lowered. For the currently available 7x7 matrix data, analyses are limited so far. In the future we might extend these by implementing drug combination response surface fits, which would help in understanding dose-dependent synergy.

For our combination screen we have collected matching day 0 viability measurements for every drug plate (n = 3,162 plates). In the recent past we have taken day 0 viability data matching a separate large set of single agent data to investigate whether these can be used to infer growth inhibition. To address this, we have calculated GR50s as described in Hafner et al.²⁷. GR50 values were highly-correlated with IC50 values, and in some cases did not detect clinically-validated biomarkers. Hence, we are seeking to understand the implications of these data in the single agent realm and have decided to not pursue this line of analysis for our drug combination data presented in this manuscript. Nevertheless, we are making matching day 0 viability measurements available to the public as part of the drug combination datasets on Figshare.

4. Recovery of known drug combinations: Several examples of known targeted therapy + chemotherapy combinations were identified. Were any existing effective chemotherapy combinations recovered (e.g., 5-FU and oxaliplatin for colorectal cancer)?

We have screened 60 pairwise combinations of chemotherapy + chemotherapy, of which two are currently in use in the clinic (source: Cancer Research UK patient information website, accessed August 2021): gemcitabine+cisplatin in multiple tissues, and gemcitabine+paclitaxel in breast/bladder. Combining gemcitabine with cisplatin showed synergy in up to 22% of breast cell lines and up to 10% of pancreas cell lines, whereas we did not see synergy for combining gemcitabine with paclitaxel in our screen.

Many chemotherapy combinations in the clinic utilise higher order combinations than the pairwise combinations screened. For example, the FOLFOX combination used to treat colon cancer is a triple combination comprising 5-FU + oxaliplatin (a pairwise combination used in our screen in pancreas), with the addition of folinic acid. Similarly, FOLFIRI (5-FU + irinotecan, another pairwise combination used in our screen, plus additional folinic acid) is also used to treat colon cancer. The quadruple combination FOLFIRINOX is used to treat colon and pancreatic cancer, again combining folinic acid with 5-FU, oxaliplatin and irinotecan. We find it interesting that these combinations include folinic acid, which is used to increase target availability in vivo, and that there are several such higher-order chemotherapy combinations where clinical benefit outweighs toxicity issues. Future screens to test these higher order combinations could be informative.

We also recognise that the clinical benefit of combinations can be conveyed through other modes of action than synergy. For instance, using clinical trials and PDX data,

Palmer and Sorger described independent drug action for combinations that confer combination benefit in the absence of additivity or synergy²⁸. While this is an interesting angle that can be explored for cell lines in the future, here we decided to focus on the strongest combination responses, i.e. synergy. We do however discuss higher-order combinations and independent drug action in the discussion section of the manuscript.

5. Potential for false positives in vitro: Were any known false positive combinations that have failed to replicate in vivo (e.g., IGF1R inhibitors + MEK inhibitors; PMID26479923) recovered by the screen? Given this risk with in vitro screens in general, does every combination need to be evaluated in vivo prior to clinical translation?

We agree with the referee that preclinical validation and potential false positive synergistic combinations are very important. The referee illustrated the example of IGF1R inhibitors paired with MEK inhibitors, which was reported to have strong synergy in 18 out of 45 (40%) CRC cell lines in vitro, but no combination response beyond the single agents in vivo²⁹. In the mentioned study the authors used primarily a combination of LW527 (IGF1R) and binimetinib (MEK). In our study we screened two IGF1R inhibitors, linsitinib and BMS-754807, in combination with the MEK inhibitor trametinib in colon. We found synergy for 3-7 cell lines (corresponding to 7-16%) of the cell lines screened, hence at a much lower level than the study of Gao et al. reported. In fact, the synergy rate observed for this target combination in our data is modest (67 colon combinations had higher synergy rates).

Nevertheless, it cannot be excluded that synergistic effects that we found in our screen might not be reproducible in in vivo follow-up studies. Hence, we strongly suggest that every combination undergoes rigorous in vitro and in vivo validation prior to clinical translation. This is stated in the discussion section.

6. Biomarker associations: Why binarize the continuous gene expression values? For high-priority hits, would a full synergy matrix of two compounds provide a more robust profile for biomarker analysis?

Gene expression was binarised for multiple reasons, but generally to facilitate discovery of biomarkers and interpretation of results. Firstly, gene expression data was Z-scored and then binarised based on z scores ≤ -2 (equivalent to significantly ‘down’) or ≥ 2 (equivalent to significantly ‘up’) to facilitate interpretation and comparisons between cell lines, particularly with regards to identification of ‘normal’ expression of a gene. Binarisation based on the Z-scores further simplifies the gene expression from a continuous value to a binary one which partially removes information but also results in

noise reduction. Binarisation also enables simple ANOVAs to be performed to discover associations with drug responses, as per the presence/absence of mutations, CNAs etc. Future work may include classical regression analysis using individual gene expression or multivariate patterns of expression to predict drug combination response. Secondly, a binary classification of elevated or low expression is more likely to be applicable to other settings than an absolute value (e.g. in vivo work or the clinic where quantitative gene expression may not be possible), hence increasing the potential utility of the biomarker associations. Binarised gene expression (high or low) is also easier to replicate in the lab, for example through overexpression or silencing techniques creating ‘high’ or ‘low’ expression, rather than a precise quantitative value.

A full matrix of two compounds could indeed be used to generate alternative synergy metrics (e.g. ZIP), which could be used as alternative inputs into our biomarker discovery pipeline. It is also true that having more measurements of a combination effect is advantageous. However, because of the scale of the primary screens and number of synergistic combinations identified, we decided instead to perform validation screens of 45 - 59 synergistic combinations in 30 cell lines per tissue. This had the advantage of enabling us to validate specific hits and provided confidence in the reproducibility of the screen as a whole. To perform this type of systematic validation as a full-matrix, while valuable, would be a large undertaking approaching ~25% of the size of the primary screen. We also note that where matching anchor and matrix data have been generated (see earlier combination of tasisib with trametinib or SCH772984), they are consistent suggesting that both approaches would yield similar biomarkers (especially for strong effects). Nonetheless, we agree that future studies could consider using a full matrix approach to refine synergy calls and biomarker analyses, particularly where effects are highly dose dependent or drug interactions are relatively weak.

7. Plots: Multiple plots are information-rich but convey limited information in print form given the small size (e.g., the drug labels in Fig. 1b and unlabeled strong outliers in Fig. 3a/Fig. 4a/Ext. Fig. 4c-d). This could be improved with labeling of additional points, addition of interactive plots online, and potentially showing important plot regions at higher magnification. It would also be helpful for the processed plot source data to be available.

We agree with the referee that some figure panels are more information-rich than others. We would like to point them towards the relevant data sources of the processed plot sources:

- Drug combination responses for all combination-cell line pairs used for heatmaps in Figures 1b and S2g-i are available for download on the *GDSC Combinations* website at <https://gdsc-combinations.depmap.sanger.ac.uk/> (username: trinity; password: dodgethis).

- Biomarker data used in the volcano plots of Figures 3a, S4b and S4c (previously S4c-d) can be found in Supplementary Table 3, and are now available to view and zoom in on in the *GDSC Combinations* website.
- For combinations in populations of unmet clinical need (basal-like breast, MSS colon and KRAS mutant colon) synergy rates and information on biomarkers and trials can be found in Supplementary Table 5. These data correspond to Figures 4a, S6a and S6b (previously S5a-b).

Furthermore, we have addressed the referee's comment by labelling additional data points of interest in Figures 3a and 4a and Extended Data Figures 4c, 6a and 6b.

8. Genetic target validation: A single siRNA pool appears to have been used to knockdown CHEK1 which raises the risk for an off-target effect. The authors did employ reagents purportedly designed to minimize off-target seed effects. How effective is the knockdown efficiency against CHEK1? Also, how much does the irinotecan IC50 shift in response to CHEK1 knockdown? This result would also be stronger if reproduced using multiple independent reagents/orthogonal methods or if rescued using a knockdown-resistant cDNA.

To address specificity of CHEK inhibition for synergistic effects of CHEKi+TOP1i we conducted two orthogonal experiments. We combined six different CHEK inhibitors selectively targeting CHEK1 (n = 2), CHEK2 (n = 1) or both (n =3) with the TOP1 inhibitor camptothecin, confirming that the effect is CHEK1 selective (Fig. 4d). In addition, we combined siRNAs against CHEK1 or CHEK2 with the TOP1 inhibitor SN-38, and demonstrated that only CHEK1 knockdown synergises (Extended Data Fig. 7a). We believe that these orthogonal approaches demonstrate the CHEK1 dependency of combination effects.

To address the specific question of knockdown efficiency of CHEK1 (and CHEK2) we have conducted two additional experiments. Firstly, we provide below Western blots confirming knock-down of CHEK1 or CHEK2 with pooled siRNAs against CHEK1 or CHEK2, respectively. We note that pooled siCHEK1 has a partial knock-down effect on CHEK2 in SW837 cells. Nonetheless, the specificity of CHEK1 knock-down and combination effects when paired with a TOP1i could be confirmed with individual siRNAs against CHEK1 (see further details below), hence confirming that combination effects are mostly driven by CHEK1 not CHEK2. These data have been included in the manuscript as Extended Data Figure 7b.

Secondly, as the previous siCHEK+SN-38 experiment used pooled siRNAs, we have repeated this experiment using four individual siRNAs against CHEK1. Western blotting confirmed knock-down of CHEK1 (plot below), and most siRNA has little/no effect on CHEK2 in SW837 cells (siCHEK1_2). Critically, the use of individual siRNAs against CHEK1 replicates the effects on viability observed for pooled siCHEK1. These data have been included in the manuscript as Extended Data Figure 7c-d.

Lastly, the referee asked whether CHEK1 knock-down leads to a shift in TOP1i IC50. To address this, we have paired pooled siRNA against CHEK1 or CHEK2 with a full dose range of SN-38 (TOP1; 0.001-9 μ M) and measured viability after 72h of drug treatment. As seen from the plots below, knock-down of CHEK1 but not CHEK2 lowers the IC50 of SN-38 by at least 7-fold compared to mock-transfected cells (SNU-81: 7.2-fold (IC50 siNT: 611.1 nM; siCHEK1: 84.4 nM; siCHEK2: 714.7 nM); SW837: 120-fold (IC50 siNT: 84.4 nM; siCHEK1: 0.69 nM; siCHEK2: 66.6 nM)). This supports the potential of using CHEK1-specific inhibitors to lower doses of TOP1 inhibitors (dose sparing) and to therefore potentially lower clinical toxicities related to application of TOP1 inhibitors. These data have been included in the manuscript as Extended Data Figure 7e.

We thank the referee for suggesting these important controls for our genetic validation of CHEK1.

9. In vivo efficacy experiments: The difference in tumor growth curves appears to be primarily driven by irinotecan treatment. The SNU-81 regrowth study is quite promising but would benefit from additional mechanistic studies. Are there any PD markers that correlate with this response? Are cell death pathways uniquely activated in vivo by the combination? Does any viable tumor tissue remain?

The referee raises important questions around in vivo efficacy and related PD markers. We have collected tumour material from LS-1034 (72h after start of drug treatment as well as the endpoint, i.e. 32 days) and SW837 (endpoint only, i.e. 24 days) xenografts, both from untreated as well as drug treated tumours. No tumour samples are available for SNU-81 xenografts, which were kept in regrowth studies. Available tumour samples were FFPE embedded, cut and IHC immunoreactivity for Ki67 (proliferation index), caspase-3 (apoptosis index) and phospho-H2AX (DNA double strand breaks) was

analysed. We analysed data for 1-3 tumours per treatment arm and 2-10 (Ki67 and phospho-H2AX) or 3-5 optical fields (caspase-3).

As seen in the plot below, 72h treatment of LS-1034 xenografts with irinotecan in combination with rabusertib leads to a significantly lower proliferation index (Ki67; left), a higher rate of cells staining positive for DNA double strand breaks (phospho-H2AX; middle) and a higher rate of apoptotic cells (caspase-3; right) compared to irinotecan monotherapy. This likely explains the more pronounced tumour growth inhibition observed in combination treated tumours. These results have been included in the manuscript in Figure 4i and as Extended Data Figure 8b.

The immediate effects at 72 hours of treatment described above are consistent (see data below) but less pronounced in tumours collected at the end of treatment (32 days for LS-1034 and 24 days for SW837). We hypothesise this more modest effect is because long-term treatment may select for drug-tolerant cells that are more proficient in repairing DNA, either because they are intrinsically so or because they have evolved mitigation strategies under drug pressure.

Collectively, our in vivo efficacy experiments support Ki67, caspase-3 and phospho-H2AX as PD biomarkers of combinatorial activity beyond single agent irinotecan activity. Moreover, since phospho-H2AX is an established biomarker of DSB formation, more pronounced positivity after combination treatment can be interpreted as a true pharmacodynamic read-out of stronger genotoxic activity, consistent with irinotecan and rabusertib combinatorial activity.

10. Compound verification: The authors state that the compound identity was confirmed by LC-MS. Was there a compound purity threshold as well? Can the compound source and purity information be reported with the resource?

Compound source is now provided in Extended Data Table 1. Compound purity is provided by the commercial vendor and is linked to each dataset because in some instances different batches of compounds were used. In addition, routine testing of compounds using LC-MS was implemented midway through screening and independent purity and compound identity confirmation data are available for 8 compounds. The

threshold applied for compound purity was 85% with a one exception where activity was confirmed based on activity profiles in known drug sensitive cell lines. Furthermore, compound activity was compared and consistent with existing independently generated monotherapy datasets (Extended Data Figure 2e).

We note that LC-MS is now part of our routine screening workflow and we have tested over 370 compound samples, confirming the identity for 98% of compounds (364 of 370 samples for 238 unique compounds), and 93% (245 of 264 samples with purity scores) of compounds exceeded our 85% compound purity threshold, indirectly supporting the identity and purity of compounds used for this study sourced from our commercial vendors.

11. GDSC2 website: The website should be made available to reviewers if considered part of the resource.

The website is now available to referees at <https://gdsc-combinations.depmap.sanger.ac.uk/> (username: trinity; password: dodgethis), and will be made public without password protection upon publication of the manuscript. Our long-term plan is to continue to upgrade the website to increase functionality.

Referee #3 (Remarks to the Author):

A well written largely descriptive manuscript documenting a monumental amount of repetitive experiments (>300,000 combination experiments) done to a high quality which would serve as a data base for researchers to use for a long time. This builds on other publications on combination and cancer done by group Nat Commun. 2019 Jun 17;10(1):2674. Conducting experiments that create a large volume of data is beneficial to the research community and gives researchers access to large amounts of data but also sophisticated analysis to go along with it. Good examples include TCGA and depmap initiatives/portals. Often such databases have gene mutation (whole genome/exome), gene expression (RNA seq), CRISPR and to a lesser extent proteomic data. The data generated in this paper will fall shy of such large data bases not because of a lesser effort of conducting such a large volume of experiments but rather a lesser diversity of output i.e. grades of synergy of two way combinations being the major output, which will be dwarfed when compared to whole genome/RNA seq/CRISPR data. The analysis done reflects what data the researchers have had available to them, i.e. 2 drug combinations and is mainly descriptive. Also the description of a CHK1 inhibitor causing synergy with irinotecan or a BCL2 inhibitor in combination with inhibitors of TOP1 does not

add significant knowledge to what is already realised in smaller hypothesis testing experiments. However a web-based portal (GDSC2) making this data accessible will be valuable to cancer researchers the world over.

We thank the referee for recognising the large number of combinations tested, the value of these large-scale datasets, and the potential impact this drug combination dataset will have for the community. We believe an additional benefit of our results compared to other valuable large-scale screening datasets (e.g. CRISPR screens) is that the results involve the use of existing drugs and are therefore more directly translatable.

Specific questions

1. What was the basis for picking the nine molecular baskets representing specific molecular subgroups (Line 274, Figure 1A)?, While markers like MSS and MSI are important biomarkers in the diseases concerned, their predictive use thus far has been in immunotherapy which was not evaluated in this manuscript.

The pan-tissue molecular baskets were chosen based on the frequency of mutated genes across the cell line panel and represent the three most frequently mutated genes. Additionally, the breast PIK3CA, basal breast, MSS colon and colon KRAS were chosen as these are populations of unmet clinical need represented within our cell line panel. We have added a sentence to the biomarker section to clarify this.

2. Small molecule drugs are known to have significant effects on multiple targets e.g. dasatinib inhibits ABL, SRC, C-KIT, SRC etc. How was this accounted for when calculating the shortest finite network distance (line 313, Figure 3b)?

We apologize for not making this clear. When calculating the shortest finite network distance, all possible pairs of targets and biomarkers are considered, and the shortest distance is reported. For example, for a single agent drug with six targets we would calculate the distance between each of the six targets and the biomarker, and report the shortest of those six distances. We have updated the text and methods section to clarify this.

3. In the discussion the authors have mentioned limitations of the data which include lack of stromal effects, impracticality of studying organoids in such high throughput experiments, which are true. Worth adding that effects on a panel of ‘normal’ non-cancer cells in selected combinations would also be of benefit as the translation of combination therapies defined pre-clinically is almost universally challenged by the inability of being able to combine the drugs without causing excessive clinical toxicity. The examples of CHK1 inhibitor + irinotecan cited as an example in the manuscript is a good case, as more than a decade of clinical research into combinations of CHK inhibitors and chemotherapy has not yielded a registration primarily due to excess normal tissue toxicity seen in clinical trials.

We appreciate the referee’s helpful suggestion regarding discussing ‘normal’, non-cancer cells as a way of estimating clinical toxicity and have now added this to the discussion.

We agree that a major hurdle for translating drug combinations to the clinic is clinical toxicity. This highlights the need for thorough preclinical validation work to be done for each combination, including profiling for sensitivity in subsets of cancer models.

As the referee states, the clinical development of CHEK1 inhibitors has been challenging, but is also illustrative of advances being made to enable the clinical use of combination therapies. A new generation of targeted inhibitors have been developed, offering improved on-target selectivity, reduced off-target activity, and consequently more manageable toxicity profiles. This can result in an increased therapeutic index. CHEK inhibitors are a good example of this: rabusertib and other isoform-selective CHEKi are now available for use in the clinic, and are well-tolerated in patients (see table below)^{11–15}. Hence, as part of the CHEK1+TOP1i follow-up work we have investigated the use of more specific CHEK inhibitors, including CHEK1 selective compounds, and have conducted all subsequent in vitro and in vivo experiments with rabusertib, which has been shown to be well tolerated in patients as a monotherapy as well as in a combination regiment³⁰.

Table of CHEK inhibitors in clinical trials and with reported outcome

Drug	Targets	Safe	Clinical activity	Max. Phase	NCTId	Reference
Prexasertib	CHEK1, CHEK2, RSK	Yes	Yes	Phase 2	NCT01115790 +16 trials (incl. 6 active trials)	13,14
SRA737	CHEK1	Yes	Yes	Phase 1/2	NCT02797964	21

GDC-0575	CHEK1	Yes	No	Phase 1	NCT01564251	11
UCN-01	CHEK1	Yes	Limited	Phase 2	NCT00072189 +4 trials	12
MK-8776	CHEK1, CDK2	Yes	NA	Phase 2	NCT01870596, NCT01521299	25
Rabusertib	CHEK1	Yes	NA	Phase 1	NCT00415636 +3 trials	30

Combinations of CHEKi and DNA damaging agents have shown to be effective in treating specific cancer types¹⁶⁻¹⁸, while their application in broader indication trials (i.e. ‘advanced solid tumours’) often led to low activity and toxicity issues^{11,19-26}. Therefore, we believe that stratification of patients to increase the therapeutic index will be key to the clinical success of CHEKi+DDA combinations. As our initial study and follow-up experiments show, CHEK1 inhibition + TOP1 inhibition have a combination benefit in the setting of MSS and KRAS-TP53 double mutant colon cancer. Furthermore, greater consideration of dosing and scheduling optimisation, incorporating an improved understanding of the factors that underpin drug efficacy and toxicity (for example by modulating Cmax and duration of exposure), could improve the therapeutic index for many drugs, whether used alone or in combination.

Overall, we believe our results represent an opportunity to refine and advance the development of selective CHEKi combinations in a defined patient population, and more broadly the ability of our screen to identify clinically meaningful combination therapies (amongst the vast number of possible combinations) for further investigation.

References

1. Meyer, C. T. *et al.* Quantifying Drug Combination Synergy along Potency and Efficacy Axes. *Cell Syst* **8**, 97–108.e16 (2019).
2. Storey, J. D. & Tibshirani, R. Statistical significance for genomewide studies. *Proc. Natl. Acad. Sci. U. S. A.* **100**, 9440–9445 (2003).
3. Orchard, S. *et al.* The MIntAct project--IntAct as a common curation platform for 11 molecular interaction databases. *Nucleic Acids Res.* **42**, D358–63 (2014).
4. Lee, H.-J. *et al.* Ras-MEK Signaling Mediates a Critical Chk1-Dependent DNA Damage Response in Cancer Cells. *Mol. Cancer Ther.* **16**, 694–704 (2017).
5. Shao, R. G. *et al.* Abrogation of an S-phase checkpoint and potentiation of camptothecin cytotoxicity by 7-hydroxystaurosporine (UCN-01) in human cancer cell lines, possibly influenced by p53 function. *Cancer Res.* **57**, 4029–4035 (1997).
6. Rydenfelt, M., Klinger, B., Klünemann, M. & Blüthgen, N. SPEED2: inferring upstream pathway activity from differential gene expression. *Nucleic Acids Res.* **48**, W307–W312 (2020).
7. Dwane, L. *et al.* Project Score database: a resource for investigating cancer cell dependencies and prioritizing therapeutic targets. *Nucleic Acids Res.* **49**, D1365–D1372 (2021).
8. Al-Ahmadie, H. *et al.* Synthetic lethality in ATM-deficient RAD50-mutant tumors underlies outlier response to cancer therapy. *Cancer Discov.* **4**, 1014–1021 (2014).
9. Tate, J. G. *et al.* COSMIC: the Catalogue Of Somatic Mutations In Cancer. *Nucleic Acids Res.* **47**, D941–D947 (2019).

10. Benedict, B. *et al.* WAPL-Dependent Repair of Damaged DNA Replication Forks Underlies Oncogene-Induced Loss of Sister Chromatid Cohesion. *Dev. Cell* **52**, 683–698.e7 (2020).
11. Italiano, A. *et al.* Phase I study of the checkpoint kinase 1 inhibitor GDC-0575 in combination with gemcitabine in patients with refractory solid tumors. *Ann. Oncol.* **29**, 1304–1311 (2018).
12. Li, T. *et al.* A phase II study of cell cycle inhibitor UCN-01 in patients with metastatic melanoma: a California Cancer Consortium trial. *Invest. New Drugs* **30**, 741–748 (2012).
13. Lee, J.-M. *et al.* Prexasertib, a cell cycle checkpoint kinase 1 and 2 inhibitor, in BRCA wild-type recurrent high-grade serous ovarian cancer: a first-in-class proof-of-concept phase 2 study. *Lancet Oncol.* **19**, 207–215 (2018).
14. Hong, D. S. *et al.* Evaluation of Prexasertib, a Checkpoint Kinase 1 Inhibitor, in a Phase Ib Study of Patients with Squamous Cell Carcinoma. *Clinical Cancer Research* vol. 24 3263–3272 (2018).
15. Hong, D. *et al.* Phase I Study of LY2606368, a Checkpoint Kinase 1 Inhibitor, in Patients With Advanced Cancer. *J. Clin. Oncol.* **34**, 1764–1771 (2016).
16. Wehler, T. *et al.* A randomized, phase 2 evaluation of the CHK1 inhibitor, LY2603618, administered in combination with pemetrexed and cisplatin in patients with advanced nonsquamous non-small cell lung cancer. *Lung Cancer* **108**, 212–216 (2017).
17. Karp, J. E. *et al.* Phase I and pharmacologic trial of cytosine arabinoside with the selective checkpoint 1 inhibitor Sch 900776 in refractory acute leukemias. *Clin. Cancer Res.* **18**, 6723–6731 (2012).

18. Marti, G. E. *et al.* Phase I trial of 7-hydroxystaurosporine and fludarabine phosphate: in vivo evidence of 7-hydroxystaurosporine induced apoptosis in chronic lymphocytic leukemia. *Leuk. Lymphoma* **52**, 2284–2292 (2011).
19. Scagliotti, G. *et al.* Phase II evaluation of LY2603618, a first-generation CHK1 inhibitor, in combination with pemetrexed in patients with advanced or metastatic non-small cell lung cancer. *Invest. New Drugs* **34**, 625–635 (2016).
20. Ho, A. L. *et al.* Phase I, open-label, dose-escalation study of AZD7762 in combination with irinotecan (irinot) in patients (pts) with advanced solid tumors. *J. Clin. Oncol.* **29**, 3033–3033 (2011).
21. Banerji, U. *et al.* A phase I/II first-in-human trial of oral SRA737 (a Chk1 inhibitor) given in combination with low-dose gemcitabine in subjects with advanced cancer. *J. Clin. Oncol.* **37**, 3095–3095 (2019).
22. Lara, P. N., Jr *et al.* The cyclin-dependent kinase inhibitor UCN-01 plus cisplatin in advanced solid tumors: a California cancer consortium phase I pharmacokinetic and molecular correlative trial. *Clin. Cancer Res.* **11**, 4444–4450 (2005).
23. Moore, K. N. *et al.* A Phase 1b Trial of Prexasertib in Combination with Standard-of-Care Agents in Advanced or Metastatic Cancer. *Target. Oncol.* **16**, 569–589 (2021).
24. Laquente, B. *et al.* A phase II study to evaluate LY2603618 in combination with gemcitabine in pancreatic cancer patients. *BMC Cancer* **17**, 137 (2017).
25. Daud, A. I. *et al.* Phase I dose-escalation trial of checkpoint kinase 1 inhibitor MK-8776 as monotherapy and in combination with gemcitabine in patients with advanced solid tumors. *J. Clin. Oncol.* **33**, 1060–1066 (2015).

26. Seto, T. *et al.* Phase I, dose-escalation study of AZD7762 alone and in combination with gemcitabine in Japanese patients with advanced solid tumours. *Cancer Chemother. Pharmacol.* **72**, 619–627 (2013).
27. Hafner, M., Niepel, M., Chung, M. & Sorger, P. K. Growth rate inhibition metrics correct for confounders in measuring sensitivity to cancer drugs. *Nat. Methods* **13**, 521–527 (2016).
28. Palmer, A. C. & Sorger, P. K. Combination Cancer Therapy Can Confer Benefit via Patient-to-Patient Variability without Drug Additivity or Synergy. *Cell* **171**, 1678–1691.e13 (2017).
29. Gao, H. *et al.* High-throughput screening using patient-derived tumor xenografts to predict clinical trial drug response. *Nat. Med.* **21**, 1318–1325 (2015).
30. Weiss, G. J. *et al.* Phase I dose-escalation study to examine the safety and tolerability of LY2603618, a checkpoint 1 kinase inhibitor, administered 1 day after pemetrexed 500 mg/m² every 21 days in patients with cancer. *Invest. New Drugs* **31**, 136–144 (2013).

Reviewer Reports on the First Revision:

Referee #1

I thank the authors for their thoughtful answers to the reviewers' questions.

The author's additional analysis addressed my first and second concerns.

My concern on the lack of understanding of why some cell lines are more sensitive to the combination of TOP1+CHEK1 was not directly addressed. The additional analysis to further refine the biomarker added value, but a more direct assessment of the author's hypothesis of DNA replication stress responsible for the additional sensitivity would be important given that the novelty of the finding is only in the discovery of markers of sensitivity to the combination.

Referee #2

The authors have substantially improved the manuscript and addressed reviewer comments via revisions to the text, additional experiments, and a detailed rebuttal. Additional details about compound source, screen methodology, and QC failures have been added. It is good to see general concordance of the 7x7 synergy matrices with the anchor dose screen calls. As the authors note, future analysis of this matrix dataset with other synergy metrics could be useful to understand any limitations of the anchor-based screening approach. I appreciate the authors' response that generating a full drug synergy matrix across many cell lines for all hit pairs would require substantial resources. The revised CHEK1 pharmacologic and genetic validation data is significantly more robust. The in vivo studies have also been strengthened by IHC studies on xenograft specimens from animals treated with the drug combination. I do think using continuous gene expression values for biomarker discovery in the future would be worthwhile. The authors have added day 0 viability measurements for community use.

The website is a great start but may require some performance optimization since pages loaded inconsistently when I viewed the site using Chrome and Safari browsers. It is interesting to browse the high-level data and the heat map summaries are intuitive. The "View combinations" table at the bottom of the page rendered inconsistently. There were multiple rendering errors listed in the Chrome console (e.g., "500 Internal Server Error: The server encountered an internal error and was unable to complete your request. Either the server is overloaded or there is an error in the application"). These issues should be addressed prior to launch. I also encourage the authors to add a way to view underlying dose response curves +/- anchor high/low drug when viewing individual combinations.

In summary, my comments have been addressed and I look forward to exploring this dataset further in the future.

Referee #3

The authors have responded to reviewers comments. They have provided appropriate explanations/clarifications and made necessary changes to the manuscript to warrant publication.

Author Rebuttals to First Revision:

Response to referees' comments

We thank the referees for their positive comments on our revised manuscript. Below in bold font are our responses to their remaining questions.

Referee #1 (Remarks to the Author):

I thank the authors for their thoughtful answers to the reviewers' questions.

The author's additional analysis addressed my first and second concerns.

My concern on the lack of understanding of why some cell lines are more sensitive to the combination of TOP1+CHEK1 was not directly addressed. The additional analysis to further refine the biomarker added value, but a more direct assessment of the author's hypothesis of DNA replication stress responsible for the additional sensitivity would be important given that the novelty of the finding is only in the discovery of markers of sensitivity to the combination.

We thank the reviewer for their positive feedback on our responses.

With respect to the third point, as the reviewer recognises, we provided further valuable details on biomarkers of response to TOP1+CHEK1 combinations, which could be important for patient stratification in any future clinical development. In addition to our observation that this combination drives cellular apoptosis, in our revised manuscript we provided new data from *in vivo* tumour xenograft studies that the combination induces enhanced genotoxic stress (as measured by phospho-H2AX induction) in tumour compared to irinotecan alone, shedding additional light on the mechanism of action. We agree that further studies in the future are warranted to more directly address the mechanism of sensitivity. However, this would require the use of a range of biochemical and cellular assays, and potentially *in vivo* studies, and we respectfully argue that these studies are beyond the scope of the existing manuscript and merit a dedicated study to adequately develop this aspect.

Referee #2 (Remarks to the Author):

The authors have substantially improved the manuscript and addressed reviewer comments via revisions to the text, additional experiments, and a detailed rebuttal. Additional details about compound source, screen methodology, and QC failures have been added. It is good to see general concordance of the 7x7 synergy matrices with the anchor dose screen calls. As the authors note, future analysis of this matrix dataset with other synergy metrics could be useful to understand any limitations of the anchor-based screening approach. I appreciate the authors' response that generating a full drug synergy matrix across many cell lines for all hit pairs would require substantial resources. The revised CHEK1 pharmacologic and genetic validation data is significantly more robust. The *in vivo* studies have also been strengthened by IHC studies on xenograft specimens from animals treated with the drug combination. I do think using continuous gene expression values for biomarker discovery in the

future would be worthwhile. The authors have added day 0 viability measurements for community use.

The website is a great start but may require some performance optimization since pages loaded inconsistently when I viewed the site using Chrome and Safari browsers. It is interesting to browse the high-level data and the heat map summaries are intuitive. The "View combinations" table at the bottom of the page rendered inconsistently. There were multiple rendering errors listed in the Chrome console (e.g., "500 Internal Server Error: The server encountered an internal error and was unable to complete your request. Either the server is overloaded or there is an error in the application"). These issues should be addressed prior to launch. I also encourage the authors to add a way to view underlying dose response curves +/- anchor high/low drug when viewing individual combinations.

In summary, my comments have been addressed and I look forward to exploring this dataset further in the future.

We thank the referee for recognising the improvements and updates we have made to the manuscript and we are pleased that their comments have been addressed.

We are committed to developing and maintaining a high-value, user-friendly website. Following the resubmission, we became aware of some unexpected problems with the website performance and apologise for these issues. To the best of our knowledge, these have already been addressed and we will continue to monitor the websites' performance in the coming weeks. In the unlikely circumstance that residual issues are identified these will be resolved in advance of publication.

We agree with the suggestion that providing ways to view the underlying dose response curves would be useful. We have a long term development plan for the website and this functionality is included in the list of enhancements. The development and testing time for the inclusion of this additional functionality is several months and will be included in future updates.

Referee #3 (Remarks to the Author):

The authors have responded to reviewers comments. They have provided appropriate explanations/clarifications and made necessary changes to the manuscript to warrant publication.

We thank Referee 3 for reviewing our responses to their comments and for supporting the manuscript's publication.